# Quantifying depressive mental states with large language models

## Abstract

Large Language Models (LLMs) may have an important role to play in mental health by facilitating the quantification of verbal expressions used to communicate emotions, feelings and thoughts. While there has been substantial and very promising work in this area, the fundamental limits are uncertain. Here, focusing on depressive symptoms, we outline and evaluate LLM performance on three critical tests. The first test evaluates LLM performance on a novel ground-truth dataset from a large human sample (n=770). This dataset is novel as it contains both standard clinically validated quantifications of depression symptoms and specific verbal descriptions of the thoughts related to each symptom by the same individual. The performance of LLMs on this richly informative data shows an upper bound on the performance in this domain, and allow us to examine the extent to which inference about symptoms generalises. Second, we test to what extent the latent structure in LLMs can capture the clinically observed patterns. We train supervised sparse auto-encoders (sSAE) to predict specific symptoms and symptom patterns within a syndrome. We find that sSAE weights can effectively modify the clinical pattern produced by the model, and thereby capture the latent structure of relevant clinical variation. Third, if LLMs correctly capture and quantify relevant mental states, then these states should respond to changes in emotional states induced by validated emotion induction interventions. We show that this holds in a third experiment with 190 participants. Overall, this work provides foundational insights into the quantification of pathological mental states with LLMs, highlighting hard limits on the requirements of the data underlying LLM-based quantification; but also suggesting LLMs show substantial conceptual alignment.

## 1 Introduction

Moods, emotions, and affective mental states more broadly, are fascinating. They guide us in major life decisions such as buying a house, choosing a partner or a career path. On the other hand, they influence our thoughts and cognition (Andrews-Hanna et al., 2022; Raffaelli et al., 2021; Bellana et al., 2022). For example, feeling joyful and happy may promote positive thinking: "I was great today", "I think they like me". Conversely, feeling depressed, worthless and suicidal may constrain the space of thoughts a person entertains to ones such as: "Life is not worth living", "I am to blame for this". In extreme cases, this may lead someone to engage in suicidal behaviours - a global issue resulting in premature deaths at a rate of 700,000 a year (WHO, 2024). Unfortunately, research to-date has mainly focused on understanding the algorithmic thought processes governing how people solve hierarchical planning problems (Correa et al., 2023b;a) or simple risky decision-making tasks (Russek et al., 2024). While some attempted to spell out the effects mood may have on such processes (Huys et al., 2012; 2015; Russek et al., 2020), these approaches are still limited in explaining the intricacies of thought and their mechanisms in psychopathology (Coppersmith et al., 2023; Millner et al., 2020).

To shed light on this problem, we take a research avenue that has seen an increased interest in evaluating the role Large Language Models (LLMs) may play in mental health. LLMs have already shown promise in modelling human behaviour and cognition across hundreds of different tasks (Binz et al., 2025), as well as neural activity on language-based tasks (Goldstein et al., 2022; Tang et al., 2023; Tuckute et al., 2024), and emotional cognition Gandhi et al. (2024). This has justified applications of LLMs to predict mental health symptoms across online texts (Xu et al., 2024)

and open-ended self-report (Hur et al., 2024), sparking methodological validation approaches using clinician-annotated datasets (Farruque et al., 2024). Importantly, there have been successful attempts to devise LLM measures of depression symptoms in the clinic and their link to ecological measures of mood and physiology (Moon et al., 2025). Clearly, LLMs hold significant promise for quantifying verbal expression related to emotions, feelings and thoughts. However, despite growing research, their fundamental limitation remain uncertain, yet are crucial for establishing effective LLM-based mental health interventions (Abdou et al., 2025; Scholich et al., 2025; Stade et al., 2024).

## 2    PROBLEM STATEMENT

As such, in this work, we focus on depressive symptoms to outline and evaluate LLMs performance on three critical tests, aiming to delineate the promises and limitations. Firstly, we evaluate five LLMs on a novel ground-truth dataset from a large online sample (770 participants), which contains standard, clinically validated quantification of depression symptoms, as well as specific open-ended descriptions of the thoughts related to each of the symptom. The performance of the LLMs (selected from Mistral, Gemma2 and Llama3 family of models) on this richly informative dataset shows an upper-bound on the performance in the domain of depression symptom quantification, allowing us to further examine the generalisation of inference about symptoms. Overall, as we show that the performance ranges from moderate to strong, we are interested in extracting a specific measure of symptoms. Indeed, this leads us to run a second test evaluating the extent to which the latent structure within the best-performing LLM captures the clinically observed pattern. We train supervised sparse auto-encoders (sSAE; Lee et al. (2025); Yun et al. (2021); Le et al. (2018)) on the hidden state representations of the open-ended symptom descriptions to predict the specific depression symptoms from verbal descriptions of thoughts. We show that selectively perturbing the sSAE weights modifies the predicted patterns of responses, further allowing us to steer the responses of the model within the ground-truth setting. Lastly, in the third test, we show that the sSAE measure responds to changes in emotional states induced by validated emotion induction experiment with 190 participants, correctly capturing the relevant mental states.

## 3    METHODS AND RESULTS

### 3.1    THE UPPER BOUND ON DEPRESSION SYMPTOM QUANTIFICATION

#### 3.1.1    GROUND TRUTH DATA COLLECTION

We recruited 770 fluent English speakers with diverse depression severity (see summary statistics in Appendix B.1.2) on Prolific (2025). [Research Ethics information blinded]. Participants provided consent and were reimbursed £8.21/h. See more details in Appendix A and B.1.

Participants' main task was to provide written responses to open-ended questions that we derived from the multiple-choice depression Patient Health Questionnaire (PHQ-9; Kroenke et al., 2001; Hur et al., 2024). See study design in Fig 1A. We created open-ended questions that corresponded to the original PHQ-9 questions. The questions target: "Interest and pleasure in activities", "Mood and feelings", "Sleep", "Energy levels", "Appetite", "Self-worth", "Focus", "Slowness/restlessness" (see Appendix B.2 for the exact phrasing of the questions - note that we removed the suicidality Question 9). Participants had 1.5 minutes to answer each open-ended question with a written response of at least 30 words. After open-ended questions, participants completed the standard PHQ-9 questionnaire, as well as two additional questionnaires assessing anxiety - Generalized Anxiety Disorder scale (GAD-7) Spitzer et al. (2006) - and depression - Self-rating Depression Scale (SDS) Zung (1965).

#### 3.1.2    LLM QUESTIONNAIRE SAMPLING

Firstly, we evaluate LLM performance on item-level multiple-choice question scores prediction given participants' item-level open-ended responses (see Fig 1B). We used the Gemma2 2b-it and 9b-it model version (GemmaTeam, 2024), as well as Llama-3.1-8B-Instruct and Llama-3.2-3B-Instruct (AI@Meta, 2024). We also used Mistral-7B-OpenOrca model (Lian et al., 2023).

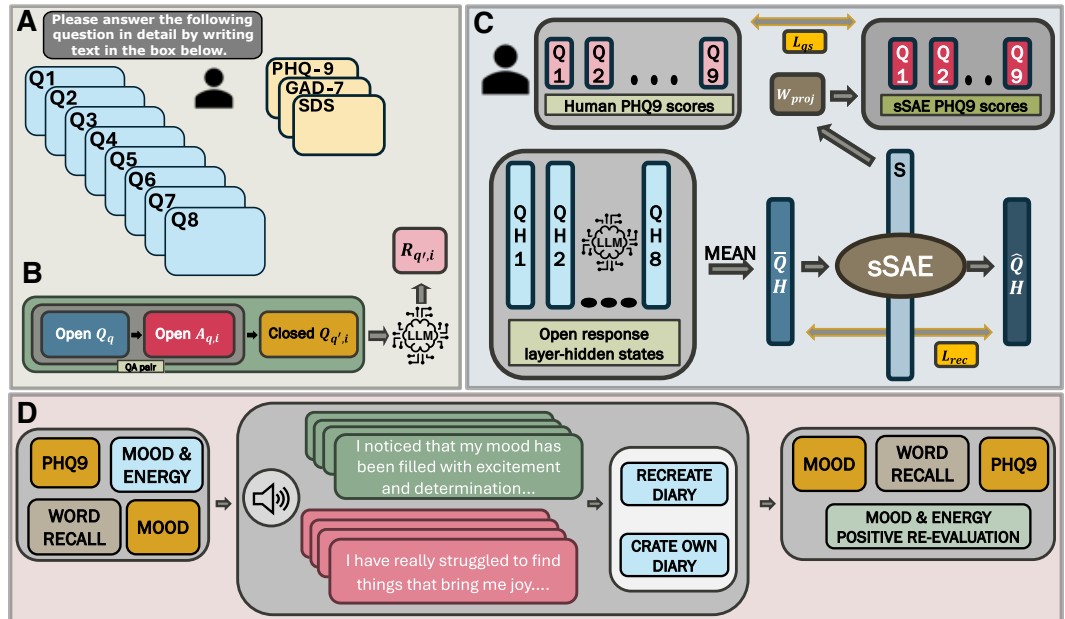

Figure 1: **A: Study 1 design.** In study 1, open-ended and equivalent multiple-choice depression symptom questions were asked. Following open-ended questions, participants completed multiple-choice PHQ-9, GAD-7 and SDS questionnaires, assessing depression and anxiety. **B: LLM prompt design and sampling**. We take the open-ended question, $OQ_q$, and participant's open-ended answer, $A_{q,i}$, forming participant's QA-pair. We append the corresponding multiple-choice question $CQ_{q'}$ and pass the prompt to an LLM to sample responses, $R_{q',i}$, on a scale. **C: Study 2 design.** We extract hidden states over participant open-ended responses and train a supervised sparse auto-encoder (sSAE) to predict PHQ-9 scores and reconstruct the original hidden state. **D: Study 3 design.** In study 3, a new group of participants completed baseline measures of: depression symptoms (PHQ-9 scores and open-ended responses about mood and energy levels), emotional word recall and momentary mood. We then allocated participants to one of positive or negative mood induction conditions. Participants listened to "diary entries" and had to first recreate what they heard to then create new diary entries that describe situations from their own life - matching the original ones as closely as possible. We then repeat the measures after mood induction to evaluate how well the latent sSAE representations of verbal descriptions track changes in the measures of participants' emotional state.

For each participant, we create a set of item-level prompts, where a prompt consists of: a QA-pair (open-ended question and participant's open-ended answer), as well as the corresponding multiple-choice question (see an example prompt in Appendix B.4). We then perform a forward pass using each of participant's QA-pair and multiple-choice question. See an illustration of this process in Fig 1B. We instruct the model to answer by selecting a character (A, B, C, D) that corresponds to the label severity response from questionnaire scale. We convert the logits across the four valid responses into probabilities and sample 50 responses for each question for each participant. We average across the samples for each QA-pair response.

In Fig 2A-H, for each question we plot subject's score on the multiple choice question against the best LLM's (Gemma2-9B) predicted score to the same question, given participant's QA-pair. Overall, predicted scores for each question given participants' corresponding QA-pair as context are moderately to highly correlated with the true participants' scores ($0.56 - 0.78$). This indicates that the LLM is able to represent the sample from participant's mental state (in the form of open-ended responses) to then provide expected responses on the multiple-choice question. We also display the other LLMs performance for each question in Fig 2I.

This emphasises the ability of the LLM to provide consistent responses across a diverse set of depression symptoms in a human population. However, there are also important discrepancies, particularly when participants report a lack of the symptom in a given question ("Not at all" answer for multiple-

choice PHQ-9 questions). This may indicate that the content of participant's answer does not give the model enough evidence to claim the lack of a symptom given that participants don't explicitly state they are not bothered by something.

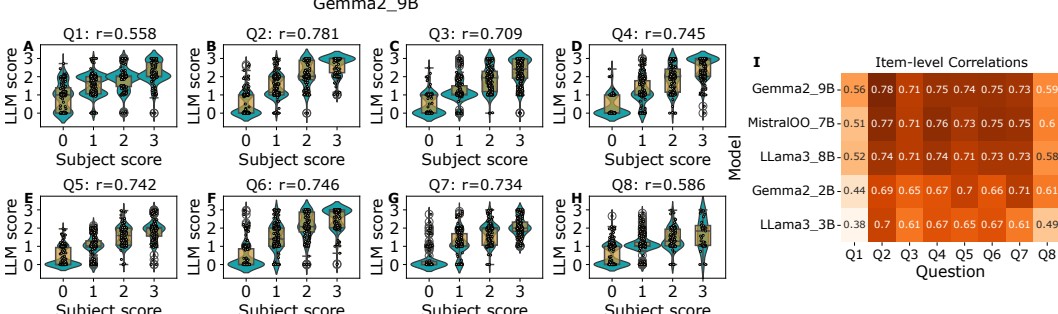

Figure 2: **A-H**: Violin, box and scatter plots, with correlations (p < 0.001) of subject's item-level scores vs the best performing model (Gemma2-9B) scores for the PHQ-9 items given the corresponding open-ended QA-pair. **I**: Correlation map showcasing performance (p < 0.001) of each LLM at predicting PHQ-9 scores from open-ended QA-pairs, see scatter plots in Appendix B.5.2.

Next, we evaluate LLMs' response generalisation to multiple-choice questions beyond those implied by QA-pairs, testing the quantification of the depressed mental state. We create a new set of item-level prompts, where a prompt consists of: one of participant's eight QA-pairs and one of all of the multiple-choice questions from the SDS, GAD-7 and PHQ-9 questionnaires. Here, we evaluate how well each LLM is able to recover multiple-choice scores given each of the QA-pairs (mental state samples in a specific dimension). We pass each of the participant's QA-pair question prompt through the model in order to sample and then average across 50 responses to the multiple-choice question.

Additionally, in Equation 1, we approximate LLMs' total score (overall symptom severity) for each of the sampled self-report questionnaire, $j$, by summing up scores, $s_{q_j,c}$, across questionnaire items, $\{q_j\}$, for each of the QA-pairs as context, $c$, and then averaging across them.

$$\hat{T}_j = \frac{1}{C} \sum_{c=1}^{C} \sum_{q_j=1}^{Q_j} s_{q_j,c} \tag{1}$$

At the item-level view, we plot the correlations between participants' scores for each PHQ-8 question (suicidality Q9 omitted) and each question from SDS questionnaire for Gemma2-9B in Fig 3A. We use this as a benchmark questionnaire response structure that participant's demonstrate and compare with the correlations between the true item score and LLM recovered score for that question given each of the open-ended QA-pairs (y-axis). We observe that the covariance structure between PHQ-8 and generalisation questionnaire scores (Fig 3A) match some of the block structure of the correlation map between the true generalisation questionnaire scores and model recovered scores (Fig 3B) from the corresponding mental state description. We also compute the average covariance items difference between the true and recovered structures for each questionnaire and model in Fig 3D.

Furthermore, in Fig 3C, we plot participants' total scores for the SDS depression questionnaire against Gemma2-9B estimated total score. The model clearly captures the expected trend in total scores (Pearson's correlation 0.84), see other questionnaires in Appendix Fig 11. We report the totals correlations for each model and questionnaire in Fig 3E.

These results are promising and indicate LLMs represents a short open-ended mental state report derived from PHQ-8 items sufficiently to generalise from this state to respond to other depression and anxiety questions in a manner reflecting the true underlying mental structure. This further indicates LLMs ability to encode a consistent representation akin to a mental state resulting in expected depression symptom responses. Importantly, these consistencies are seen both at an item and questionnaire level. Overall, these findings establish an upper bound on depression symptom quantification with LLMs, allowing to examine how the performance generalises beyond this domain.

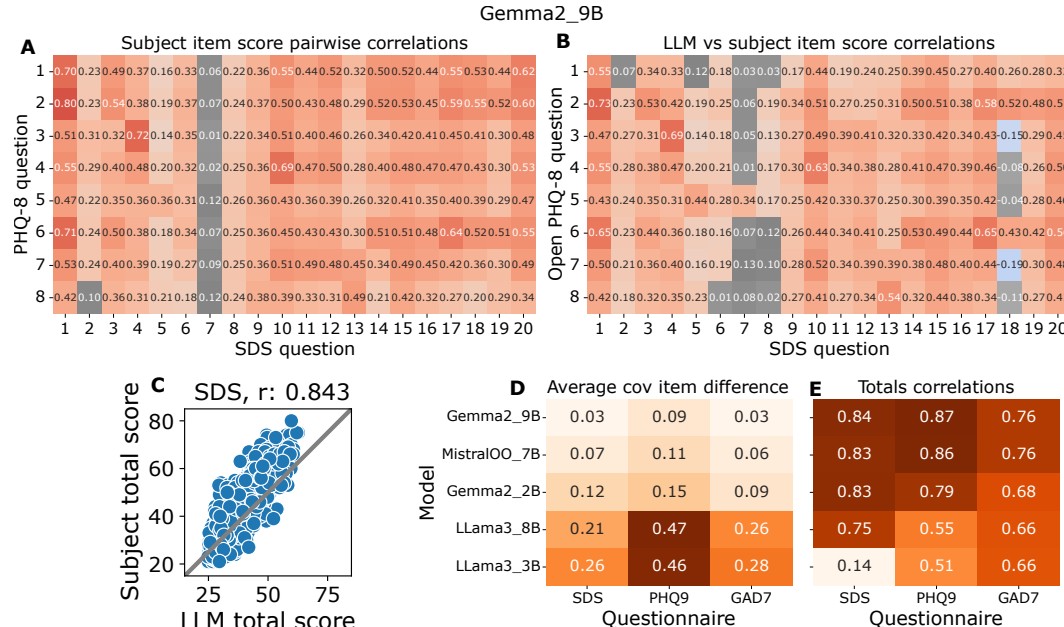

Figure 3: **A**: Subjects' ground-truth pairwise correlations between PHQ-8 item scores and SDS scores. **B**: Correlations between true SDS item scores (x-axis) and Gemma2-9B recovered scores for that question given each of the open-ended PHQ-9 QA-pairs (y-axis). Significant correlations are in cool-warm colour. The correlations not reaching significance are in gray scale. **C**: Subject's total scores for the SDS depression questionnaire against the best model's (Gemma2-9B) total score estimate (Pearson correlation reported). **D**: For each LLM and questionnaire (SDS, PHQ9, GAD7), we plot the average difference between the ground-truth and model-recovered correlation-matrix items. **E**: Correlation between model total score estimates and true total scores for each LLM and questionnaire. Detailed results for other questionnaires and models are reported in Appendix B.5.3.

## 3.2 QUANTIFYING THE LATENT STRUCTURE OF DEPRESSION SYMPTOMS

Having establish the upper-bound in depression symptom prediction in the first test case, we are now interested in extracting a specific measure that encompasses this. We expand on sampling results, and ask to what extent the latent structure within LLMs hidden states captures the observed pattern.

### 3.2.1 SUPERVISED SPARSE AUTO-ENCODER (sSAE)

We use the ground truth dataset collected in Study 1 to train a supervised sparse auto-encoder (sSAE; Lee et al. (2025); Yun et al. (2021); Le et al. (2018)) to predict participants' z-scored PHQ-9 scores from the hidden states representations of the QA-pairs of open-ended prompt. We extracted hidden-states on the token just after the open-ended response for each layer of the model from the middle layer onwards. We averaged the hidden states across eight questions obtaining an average hidden state for each participant, $h_i$, pairing this with the true vector of PHQ-9 scores $y_i$. We report the sSAE model Equations 2-5 below:

$$h_{cent} = h - b_{dec} \tag{2}$$

$$s_{post} = relu(h_{cent}W_{enc}) \tag{3}$$

$$q = s_{post}W_{proj} + b_{proj} \tag{4}$$

$$h_{rec} = s_{post}W_{enc}^T + b_{dec} \tag{5}$$

During training, the goal is to both reconstruct the original hidden state $h_i$ ($l_{rec} = ||(h_{rec}-h)||^2$) as well as predict the vector of scores $y_i$ by projecting from the latent sSAE latent state $s_{post}$ to latent PHQ-9 score $q_i$ ($l_{qs} = ||(q-y)||^2$). See the details of architecture, training and hyper-parameter tuning in Appendix C.1.

We focused on the best performing LLM from Study 1 (Gemma2-9B). We first plot the sSAE PHQ-9 item-level predictions for the best layer in Fig 4A-I, as well as the predictions correlations for each layer in Fig 4J. We observe that the performance varies between questions, mostly capturing the overall severity trend. For some questions, the performance approaches that of the sampling method in study 1, indicating that the sSAE representations capture the relevant depression symptom structure. Interestingly, the prediction of suicidality question 9, increases over the layers, perhaps showcasing the necessity to better integrate relevant information for this out-of-sample prediction.

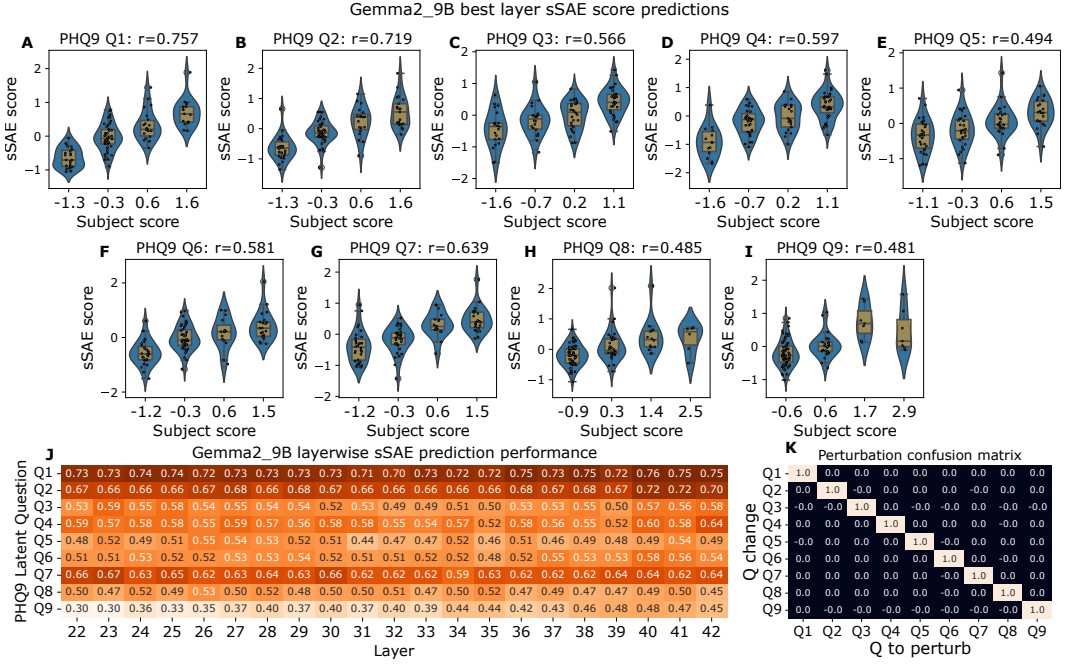

Figure 4: **A-I**: Violin, box and scatter plots, with correlations (p < 0.001) of subject's item-level z-scores vs the best performing sSAE model (layer 42, Gemma2-9B) predicted scores for PHQ-9 items, given the average hidden state across all open-ended responses for each participant. **J**: Correlation between participants' item-level scores and sSAE predicted scores for each layer and each question. **K**: Confusion matrix of latent sSAE score perturbation sensitivity analyses. For each target question we seek a change in the sSAE sparse representation to increase latent PHQ-9 score by 1. The diagonal matrix of latent changes implies the perturbation is selective to each question.

### 3.2.2 LATENT STATE PERTURBATION SENSITIVITY

To ascertain the link between sSAE latent representations and the PHQ-9 predictions, we evaluate the sensitivity of the measure to slight item-level perturbations. We aim to achieve a specific change in question $\Delta q_j = [0, .., \delta_j, ..., 0]$, such that $q_{new,j} = q + \Delta q_j$. Therefore, following the procedure specified in Appendix C.1.3, we seek a change, $\Delta s_{post,j}$, to the latent sSAE state where:

$$s_{new,j} = s_{post} + \Delta s_{post,j} \qquad (6)$$

For each question and layer separately, solving the basis pursuit problem (Van Den Berg & Friedlander, 2009), we find the $\Delta s_{post,j}$ to project to the new latent sSAE PHQ-9 score:

$$q_{new} = s_{new,j} W_{proj} + b_{proj} \qquad (7)$$

We plot the average difference between the original $q$ and perturbed $q_{new}$ across held-out participants for each pair of questions as a confusion matrix in Fig 4K, showcasing the item-level specificity of the perturbation.

### 3.2.3 STEERING SYMPTOM RESPONSES WITH SSAE

Lastly, we seek to establish the effect of the specific latent perturbations in the sSAE space on responses in the item-level questionnaire sampling. We repeat the procedure as in Study 1 - for

each QA-pair's open-ended response we extract its hidden states, averaged with that of the last token in the prompt. We pass this averaged hidden state through the sSAE to then perturb the latent, sparse representation of each question using the method above, both in positive and negative severity direction. We move the original hidden state towards the reconstructed, perturbed hidden state so as to bias the logits of the questionnaire responses. See the exact steering implementation details in Appendix C.1.4.

In Fig 5, we find that we are able to steer most of the question responses successfully, both in the negative (top row, less symptom severity) and positive (bottom row, more symptom severity) direction. The strength of this perturbation varies between questions, but nevertheless demonstrates the sensitivity of the latent sSAE measure.

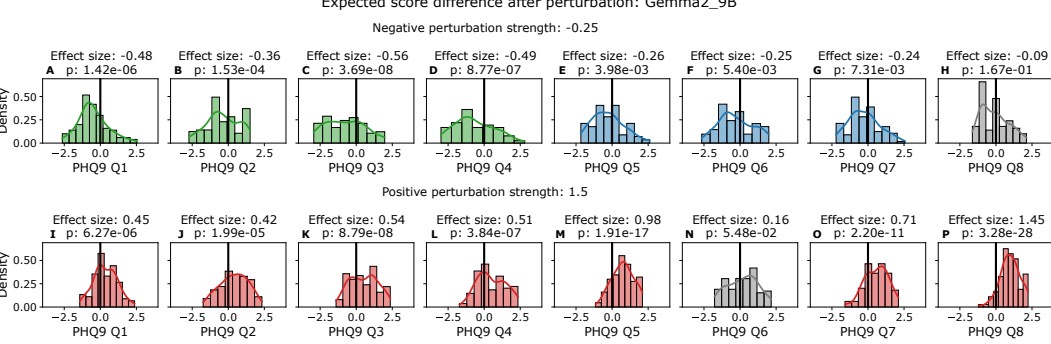

Figure 5: The effect of hidden states sSAE perturbation on sampled item-level PHQ-9 responses, given participants' corresponding open-ended QA-pair. We plot histograms of differences between steered sampled scores and original scores for **A-H**: positive direction (less symptom severity) and **I-P** negative direction (more symptom severity). For each, we report the Cohen-d effect size and the p-value of one-sided t-test. We use blue for small effects $|d| < 0.3$, green for positive direction effects $d < -0.3$ and red for negative direction effects $d > 0.3$.

Overall, using the sSAE representations, we find that we can extract a symptom-meaningful latent structure from the LLM hidden state. As demonstrated through perturbation analyses of these states, sSAE provides a sensitive and specific measure of the observed symptom patterns.

### 3.3 LLM LATENT STRUCTURE IS SENSITIVE TO HUMAN EMOTIONAL STATE CHANGES

Lastly, in the third test case of establishing the limits of depression symptom quantification with LLMs, we evaluate to what extent the sSAE measure responds to changes in human emotional states as induced by emotion induction intervention.

#### 3.3.1 DATA COLLECTION

We recruited 190 fluent English speakers with diverse depression severity (see summary statistics in Appendix D.1.2) on Prolific (2025). [Research Ethics information blinded]. Participants provided consent and were reimbursed £8.21/h. See more details in Appendix A and D.1. Participants completed baseline measures that consisted of depression symptoms (PHQ-9 scores on a continuos visual analogue scale (VAS), and open-ended responses about mood and energy levels), sentiment of emotional word recall and momentary mood rating ("How happy are you at this moment?") on a VAS, see details in Appendix D.2.

We then allocated participants to one of positive (n=97, Mood High (MH)) or negative (n=93, Mood Low (MH)) mood induction conditions. In each condition, participants listened to "diary entries" (see Appendix D.3 for intervention design and transcripts) and had to first recreate by typing, one word at a time, what they heard. They then had to create new, autobiographical diary entries describing situations from their own life - matching the original ones as closely as possible. See exact instructions in Appendix D.4. Importantly, the diary entries are created based on the open-ended responses from Study 1, aiming to induce states we already quantified with LLMs before. After mood induction, we took the same measures as at baseline, but this time we asked participants to

re-evaluate their original open-ended responses about mood and energy levels in a more positive light. See the depiction of the study design in Fig 1D

### 3.3.2 COGNITIVE MEASURES

We focus on the following cognitive measures that track the emotional state of participants. In particular, we compare the post-mood-induction (FU) to baseline measures. We compute: the difference in PHQ-9 question 2 score ("Feeling down, depressed, or hopeless"), the difference in momentary mood rating and the difference in the average sentiment of the recalled words. We plot the measures in Fig 6A. We find that all the change measures are significantly different between conditions (PHQ-9 Q2: $t(188) = 2.195$, $p = 0.029$; mood: $t(183) = -6.314$, $p < 0.001$; recall sentiment: $t(188) = -2.770$, $p = 0.006$). This shows we successfully induced the desired emotional states.

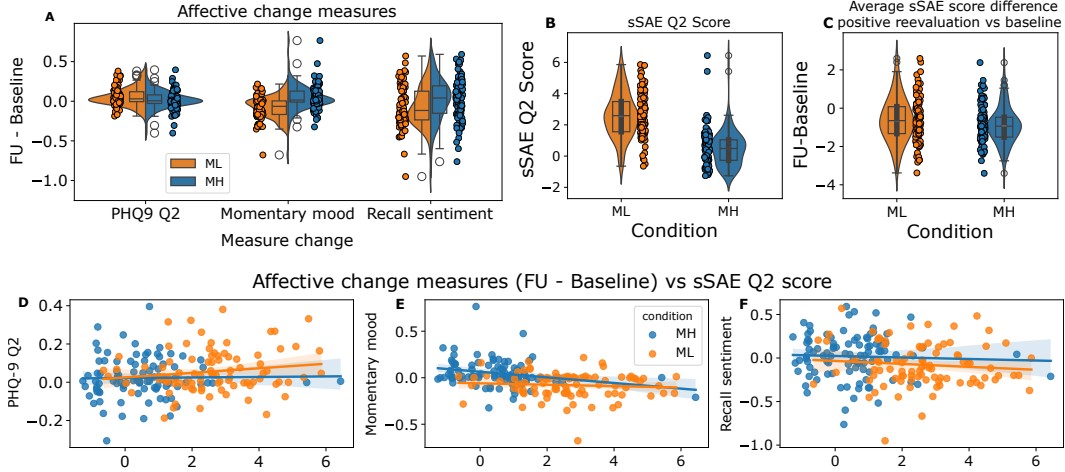

Figure 6: **A**: Violin, box and scatter plots of changes in cognitive measures after mood-induction (FU) for each condition (negative ML - mood low; positive MH - high mood) relative to baseline. Interventions are effective as the differences for PHQ-9 Q2 score, momentary mood and word recall sentiment show significant differences between conditions. **B**: sSAE latent Q2 scores for each participant's created, autobiographical diary entry show the sSAE measure significantly differentiates the inducing emotional state of each condition. **C**: Average sSAE latent scores changes between open-ended baseline mood and energy evaluation and FU positive re-evaluations of these symptoms may suggest (p= 0.081) that participants in the negative mood induction find it more difficult to get out of the negative mindset. **D-F**: Pulling across conditions, sSAE latent Q2 score capture the overall trend in the emotional change measures (participants' PHQ9 Q2 score, momentary mood and recall sentiment).

### 3.3.3 SSAE MEASURES OF EMOTIONAL STATE

Following the effective emotional state induction, we then asked whether the sSAE measures derived in Study 2, track these emotional changes. To that end, we took participants created, autobiographical diary entries and obtained hidden states representations from the Gemma2-9B model at the last token for each layer. We applied the layer-wise sSAE to calculate latent PHQ-9 scores from the diary texts. Indeed, in Fig 6B, we show that the layer-average ssAE Q2 measure is sensitive to these described emotional states ($t(185) = 10.783$, $p < 0.001$, see scores for other texts (baseline and positive re-evaluation of mood and energy levels symptoms) in Appendix D.5.1).

Additionally, we look at the FU and baseline open-ended mood and energy evaluations and compute the time-point difference in Fig 6C, for the average latent score across all questions and layers. We find a trend-level result, suggesting participants in the negative mood induction condition find it more difficult to positively re-evaluate themselves compared to those in the positive mood induction ($t(186) = 1.755$, $p = 0.081$). This may capture previous findings suggesting that depressed individuals find it more difficult to get out of the negative mindset (Andrews-Hanna et al., 2022).

Lastly, in Fig 6D-F, we show that the sSAE Q2 score captures the overall trend in the above-mentioned measures of emotional state change, although this is not condition-specific (PHQ9 Q2: $t(185) = 2.279$, $p = 0.024$; mood: $t(181) = -5.538$, $p < 0.001$; recall sentiment: $t(185) = -2.356$, $p = 0.020$)

Taken together, we show that LLM latent measures capture and quantify relevant mental states from texts that induce specific emotional changes, further establishing the sensitivity and limits of LLM quantifications.

## 4 DISCUSSION AND CONCLUSION

This work provides foundational insights into the quantification of pathological mental states with LLMs. Specifically, we established three critical tests to evaluate the performance of LLMs on capturing and measuring depression symptoms. Firstly, using the novel ground-truth dataset of 770 participants, we established that LLMs perform well on reproducing depression symptoms patterns from open-ended descriptions of related thoughts, with some variation between models. Importantly, we show that this generalises to inference about different symptom profiles, reflecting the response patterns observed across the population. This first test provides an upper bound on the performance in this domain obtained through the comparison against the ground-truth in our dataset. This lead us to perform the second test to evaluate the measures and predictions of symptoms obtained with the supervised sparse auto-encoder (sSAE) approach, quantifying the relevant latent LLM structure (hidden states). We show that sSAE measures pass the second test, both by reproducing the response patterns, but also by offering a way to perturb the latent state so as to steer responses in a desired direction. Lastly, in a separate experiment, we show that the sSAE measures pass the third test by tracking the changes in relevant mental states induced by emotional induction intervention across 190 participants.

There are some limitation to our study. Firstly, in Study 1, we only focused on three self-report questionnaires (SDS, GAD-7, PHQ-9). We also used mental state descriptions based on a specific, structured PHQ-9 questionnaire. It's unclear how well the LLM would perform with other questionnaires and if the findings would generalise to other types of assessments. However, an existing body of work, showcase the capability of LLMs to represent human-like personality profiles (Rutinowski et al., 2023; Serapio-García et al., 2023), as well as mental states relating to psychopathology Coda-Forno et al. (2024) and more broadly (Hagendorff et al., 2024; Jiang et al., 2024), especially when meeting some of the established criteria for using psychological tests on LLMs (Löhn et al., 2024; Huang et al., 2024), Given these results, we expect the LLMs to perform equally well on different domains of psychopathology.

Secondly, we acknowledge that the quality and informativeness of open-ended responses may have varied across participants (especially the slow-typers) due to 1.5 minutes time-limit and a minimum 30-words requirement. We also didn't control for confounding variables such as current mood due to the most recent events, which may not be informative about a more persistent depressive state. In Study 3, we only induced one specific emotional state dimensions, and so cannot ascertain how sensitive the other sSAE dimensions are. Furthermore, we recruited a specific online population on Prolfic, selecting for those with depressive symptoms. Further analyses would have to be performed to establish whether LLMs are able to quantify other mental states across a diverse subsets of populations, including healthy controls as well as participants in lab environments.

Lastly, while the performance may vary between models and while the sSAE latent measures offer less than perfect prediction and control over the symptom responses, nevertheless, LLMs provide a way to quantify depression symptoms from verbal descriptions. Our three-fold framework provides a way to evaluate this, allowing to highlight the limitations and requirements any LLM-measure of symptoms should meet so as to be of good use in the field of mental health (mental state assessment, therapeutic intervention allocation and delivery). Importantly, the third test offers a promise of deriving validated, symptom-grounded measures from any text, a notable use in cases where identifying those at higher risk of suicide may be obscured due to stigma or fear (Hallford et al., 2023).

## 5 ETHICAL CONSIDERATION

Given the sensitive nature of the questions asked, we took necessary measures to ensure participants are comfortable in answering the questions and provided them with mental health resources. Participants were also given the option to withdraw at any point. We see our research to have a potentially high positive societal impact, aiming at mental health assessment and improved, personalised treatment. The translation of our research could be applied as scale for example by integrating with mental health chatbots or with smartphone ambulatory data, helping as many as possible.

The study was approved by the authors' institutional (university) Research Ethics board [Research Ethics information blinded]. Our ethical approval allows for indefinite storage of collected data on university servers and on password-protected, encrypted personal computers accessible only to select few individuals in the research group. We are also compliant with the Data Protection Act 2018 and GDPR. As participants are free to withdraw at any point (including after data collection), we are obliged to delete their data – unless it's already published. More generally, if we find that it's impossible to say that there is no identifiable information, we won't make the dataset publicly available, but will retain the data for research purposes, in line with our ethical approval.

As such and due to the sensitive nature of the collected data we won't be releasing the dataset publicly. However, we will consider access requests from verified researchers as well as sharing of the data in the collaborative agreement capacity.

## 6 REPRODUCIBILITY STATEMENT

We attached a .zip file of the code repository for reference, which includes the code used through the study as well as random seeds for torch. This is for reference only as the scripts require data, which we cannot provide due to ethical considerations. We report python packages and versions in the Appendix E.1. We describe the data collection procedures, inclusion/exclusion criteria for the participants studies, the exact content of the instructions and questions asked, as well as transcripts of the emotional induction interventions in the Appendix B, D. We also list the clinically standardised list of symptoms asked in the Appendix B.3. We list LLMs used in the study in the Appendix E.2. Furthermore, we provide the details of questionnaire LLM sampling prompt in Appendix B.4, and the details of hyper-parameter grid search and layers used in the sSAE analyses C.1. This also includes the perturbation methods and strength search performed during the sSAE perturbation analyses. Lastly, we used the same train, validation and test set from the from test throughout the study. Any hyper-parameter search was performed on the validation set, while the final performance was evaluated against the held-out test set.

## 7 THE USE OF LARGE LANGUAGE MODELS (LLMs)

We used LLMs in places to help rephrase the manuscript text, as well as rephrase some of the mood induction texts used in Study 3. At points, we used LLMs to debug and suggest code snippets for certain data processing and analysis steps.

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

# APPENDIX

## A  PARTICIPANT RECRUITMENT

The participants studies were approved by [Research Ethics information blinded]. All participants (18 or above) provided online consent after reading provided Participant Information Sheet. Participants were informed that they would be asked questions about their mood and feelings and we have provided information about ways to seek help should they feel affected by the issues raised by these questions. We reimbursed participants at a rate of £8.21/h, which as rate approved by the Ethics Committee (appropriate for the demographic). For the analyses, we have pseudo-anonymised the data (changed the Prolific ID, which in itself ensures participants anonymity), but cannot ensure no identifiable is contained in the open-ended responses. Participants were free to withdraw at any point, and we have emphasised that throughout the study.

We recruited a sample of participants from Prolific (2025) to represent diverse range of depressive symptoms. We screened for Prolific participants who were willing to participate in a study about sensitive topics (mental health, emotions, feelings) and potentially harmful content, who had approval rate of 95-100%, at least 5 previous submissions.

## B  QUANTIFYING OPEN-ENDED DEPRESSION SYMPTOMS - STUDY 1

### B.1  PARTICIPANT RECRUITMENT

Participants were instructed that they will have to provide written responses as well as multiple-choice responses. They answered one open-ended question at a time within 1.5 minutes, while the multiple-choice questions were displayed on one scrollable page.

#### B.1.1  INCLUSION/EXCLUSION CRITERIA

We excluded participants who meet at least one of the criteria:

- timed out or provided less than 30-word response to any of the questions more than 4 times in total,
- failed two attention checks.

In total we included 770 participants.

#### B.1.2  DEPRESSION AND ANXIETY SEVERITY

We reported the summary statistics of total scores for PHQ-9, GAD-7, SDS below:

- PHQ-9 total $11.1 \pm 7.11$ (depression)
- SDS total $48.13 \pm 12.43$ (depression)
- GAD-7 total $8.81 \pm 6.22$ (anxiety)

### B.2  OPEN-ENDED QUESTIONS

1. Could you share any activities or events from the past two weeks that made you feel bothered because of a lack of interest or pleasure in doing them?

2. For the past two weeks, have you been bothered by your mood and how you felt generally? Were there any situations when you felt down, depressed, or hopeless?

3. Can you provide examples of how your sleep has been in the past two weeks? Have you been bothered by challenges with falling asleep, staying asleep, or even sleeping too much?

4. Have you been bothered by your energy levels over the past two weeks? Can you recall situations when it comes to feeling tired/lively or low/high on energy?

5. In the past two weeks, have you been bothered about your appetite? Can you describe your typical attitude towards food - maybe you have noticed something unusual, like changes in how much you're eating or not eating?

6. In the past two weeks, have you been bothered by feelings about yourself? In what situations did you feel proud or like a failure? Did you feel you met your own and your family's expectations, or let them down?

7. In the past two weeks, have you been bothered by your ability to concentrate and focus? Please describe how it felt to do things that require you to concentrate for a while, like working, reading, or watching movies?

8. Can you describe situations over the past two weeks when you were bothered by feeling slower than usual in terms of thinking, speaking, or just acting - or situations where you felt fidgety and restless?

## B.3 MULTIPLE-CHOICE QUESTIONS

### B.3.1 PHQ-9 ITEMS

Participants were instructed to consider "Over the last 2 weeks, how often have you been bothered by any of the following problems?". These are answered on a scale - Not at all (0), Several days (1), More than half the days (2), Nearly every day (3) (Kroenke et al., 2001).

1. Little interest or pleasure in doing things.

2. Feeling down, depressed, or hopeless.

3. Trouble falling or staying asleep, or sleeping too much.

4. Feeling tired or having little energy

5. Poor appetite or overeating.

6. Feeling bad about yourself - or that you are a failure or have let yourself or your family down.

7. Trouble concentrating on things, such as reading the newspaper or watching television.

8. Moving or speaking so slowly that other people could have noticed? Or the opposite - being so fidgety or restless that you have been moving around a lot more than usual.

9. Thoughts that you would be better off dead or of hurting yourself in some way.

### B.3.2 SDS ITEMS

These are answered on a scale - A little of the time (1), Some of the time (2), Good part of the time (3), Most of time (4). Questions 2, 5, 6, 11, 12, 14, 16, 17, 18 and 20 are reverse scored Zung (1965).

1. I feel down-hearted and blue

2. Morning is when I feel the best

3. I have crying spells or feel like it

4. I have trouble sleeping at night

5. I eat as much as I used to

6. I still enjoy sex

7. I notice that I am losing weight

8. I have trouble with constipation

9. My heart beats faster than usual

10. I get tired for no reason

11. My mind is as clear as it used to be

12. I find it easy to do the things I used to

13. I am restless and can't keep still

14. I feel hopeful about the future

15. I am more irritable than usual

16. I find it easy to make decisions

17. I feel that I am useful and needed

18. My life is pretty full

19. I feel that others would be better off if I were dead

20. I still enjoy the things I used to do

### B.3.3  GAD-7 ITEMS

Participants were instructed to consider "Over the last 2 weeks, how often have you been bothered by any of the following problems?". These are answered on a scale - Not at all (0), Several days (1), More than half the days (2), Nearly every day (3) (Spitzer et al., 2006).

1. Feeling nervous, anxious or on edge

2. Not being able to stop or control worrying

3. Worrying too much about different things

4. Trouble relaxing

5. Being so restless that it is hard to sit still

6. Becoming easily annoyed or irritated

7. Feeling afraid as if something awful might happen

### B.4  EXAMPLE ITEM-LEVEL LLM PROMPT

We used a system prompt "You are a human participant in a study." Below is the prompt we used to sample responses from the logits of each response label character (A, B, C, D).

```
<start_of_turn>user
Please answer the following question in detail.

Question:
For the past two weeks, have you been bothered by your mood and how you
    felt generally? Were there any situations when you felt down,
    depressed, or hopeless?

Answer:<end_of_turn>
<start_of_turn>model
 *** Participant's response redacted*** <end_of_turn>
<start_of_turn>user
Please answer the following question.

Over the last 2 weeks, how often have you been bothered the following
    problem?

Problem: Feeling down, depressed, or hopeless.

Please answer by responding only with one letter (A, B, C, D) from the
    scale below:
A - Not at all
B - Several days
C - More than half the days
D - Nearly every day

Answer:<end_of_turn>
<start_of_turn>model
```

## B.5 LLM SAMPLING

### B.5.1 LOGIT SAMPLING

We use participants' open-ended question and answer as context for the LLM to responses to the corresponding multiple-choice question on the scale (see prompt in B.4 We specify that the model has to respond by selecting a character label (A, B, C, D) that corresponds to the questionnaire label response. We do a forward pass through the model to extract the logits at the final token and then select logits corresponding to the (A, B, C, D) tokens. We then covert this restricted logits space into probability using softmax transformation. We then sample 50 responses for each question, each participant.

### B.5.2 ITEM LEVEL SAMPLING - REMAINING MODELS MODELS RESULTS

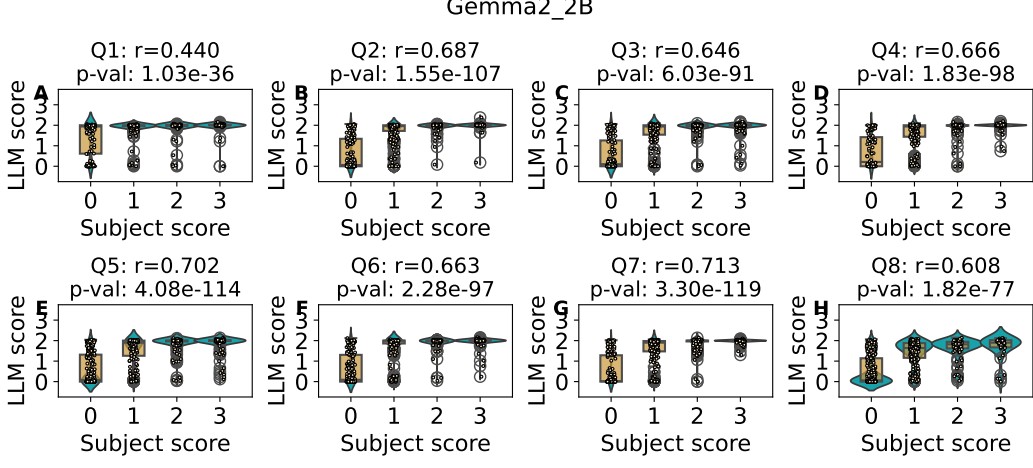

Figure 7: **A-H**: Violin, box and scatter plots, with correlations (p < 0.001) of subject's item-level scores vs the best performing model (Mistral-7B-OpenOrca) scores for the PHQ-9 items given the corresponding open-ended QA-pair.

Figure 8: **A-H**: Violin, box and scatter plots, with correlations (p < 0.001) of subject's item-level scores vs the best performing model (Gemma2-2B) scores for the PHQ-9 items given the corresponding open-ended QA-pair.

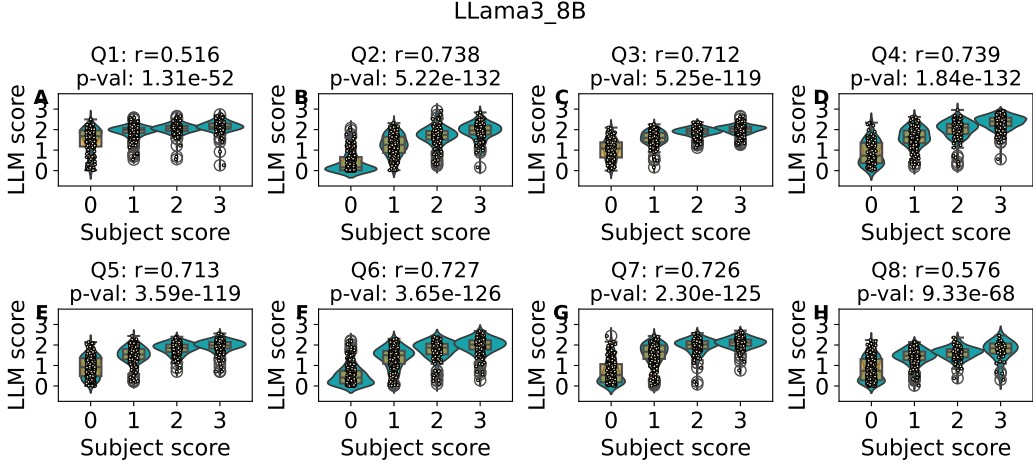

Figure 9: **A-H**: Violin, box and scatter plots, with correlations (p < 0.001) of subject's item-level scores vs the best performing model (Llama3.1-8B) scores for the PHQ-9 items given the corresponding open-ended QA-pair.

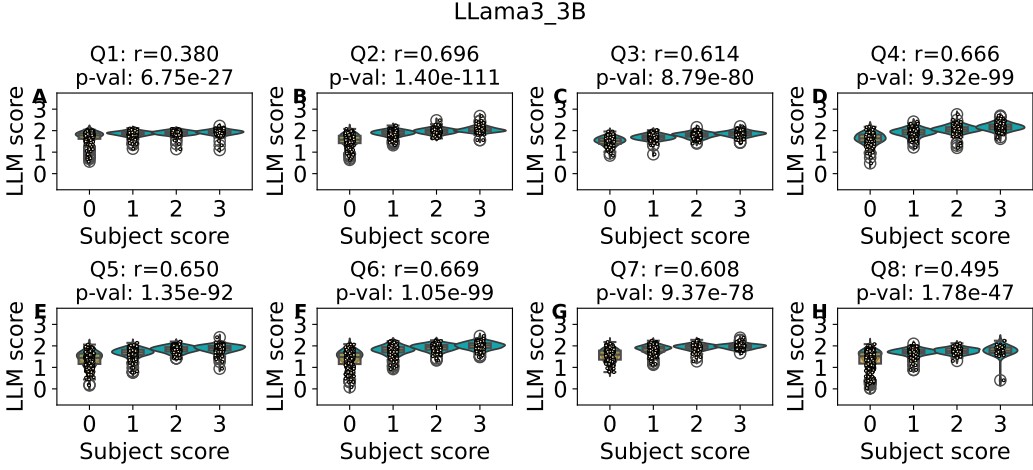

Figure 10: **A-H**: Violin, box and scatter plots, with correlations (p < 0.001) of subject's item-level scores vs the best performing model (Llama3.2-3B) scores for the PHQ-9 items given the corresponding open-ended QA-pair.

### B.5.3 QUESTIONNAIRE GENERALISATION LOGIT SAMPLING –REMAINING MODELS RESULTS

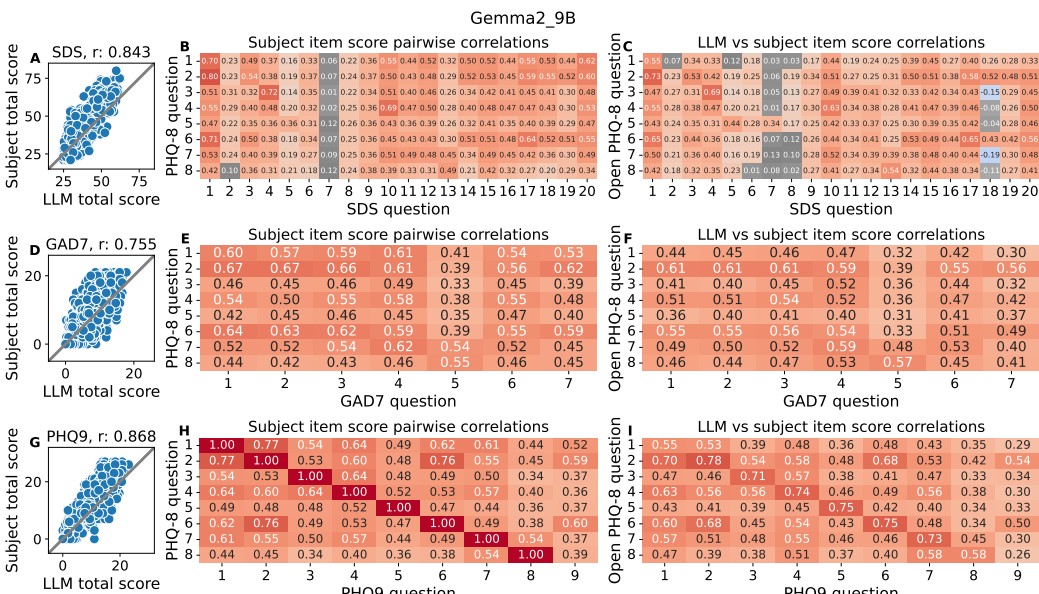

Figure 11: *Column I* (**A, D, G**): Subject's total scores for each questionnaire against the Gemma2-9B total score estimate (Pearson correlation reported). *Column II* (**B, E, H**): Subjects' pairwise correlations between PHQ-9 item scores and SDS, GAD-7 and PHQ-9 scores. *Column III* (**C, F, I**): Correlations between participant's true item score for each questionnaire (x-axis) and Gemma2-9B recovered score for that question given each of the open-ended PHQ-9 QA-pairs (y-axis). Significant correlations are in cool-warm colour. The correlations not reaching significance are in gray scale.

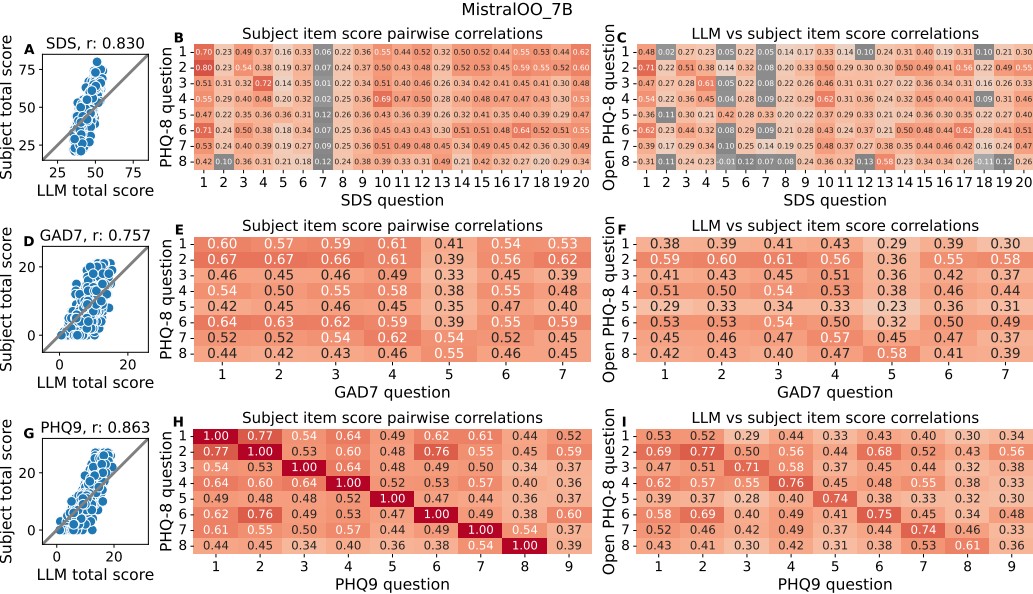

Figure 12: *Column I* (**A, D, G**): Subject's total scores for each questionnaire against the Mistral-7B-OpenOrca total score estimate (Pearson correlation reported). *Column II* (**B, E, H**): Subjects' pairwise correlations between PHQ-9 item scores and SDS, GAD-7 and PHQ-9 scores. *Column III* (**C, F, I**): Correlations between participant's true item score for each questionnaire (x-axis) and Mistral-7B-OpenOrca recovered score for that question given each of the open-ended PHQ-9 QA-pairs (y-axis). Significant correlations are in cool-warm colour. The correlations not reaching significance are in gray scale.

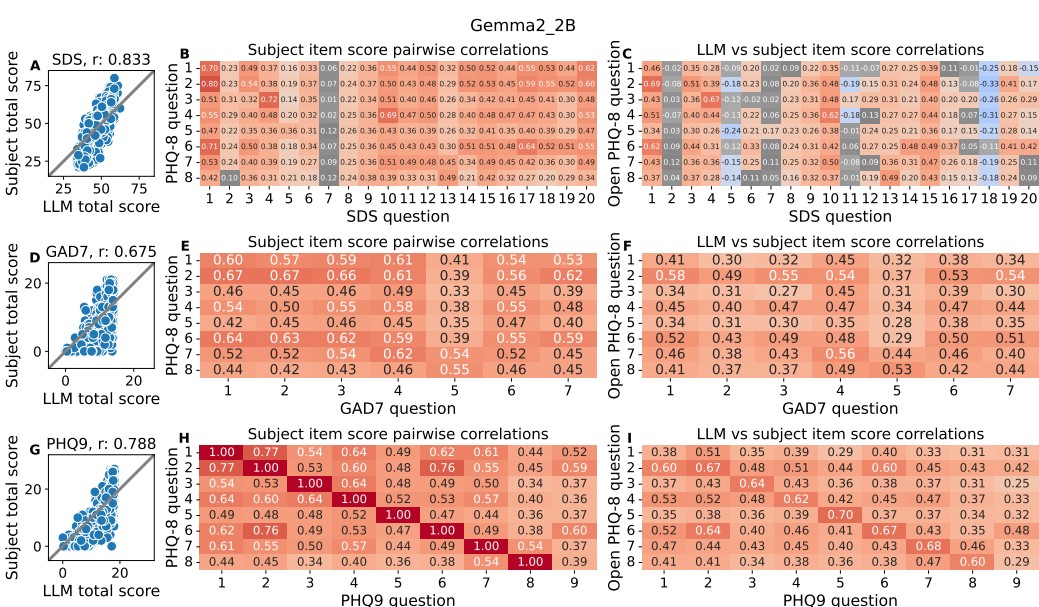

Figure 13: *Column I* (**A, D, G**): Subject's total scores for each questionnaire against the Gemma2-2B total score estimate (Pearson correlation reported). *Column II* (**B, E, H**): Subjects' pairwise correlations between PHQ-9 item scores and SDS, GAD-7 and PHQ-9 scores. *Column III* (**C, F, I**): Correlations between participant's true item score for each questionnaire (x-axis) and Gemma2-2B recovered score for that question given each of the open-ended PHQ-9 QA-pairs (y-axis). Significant correlations are in cool-warm colour. The correlations not reaching significance are in gray scale.

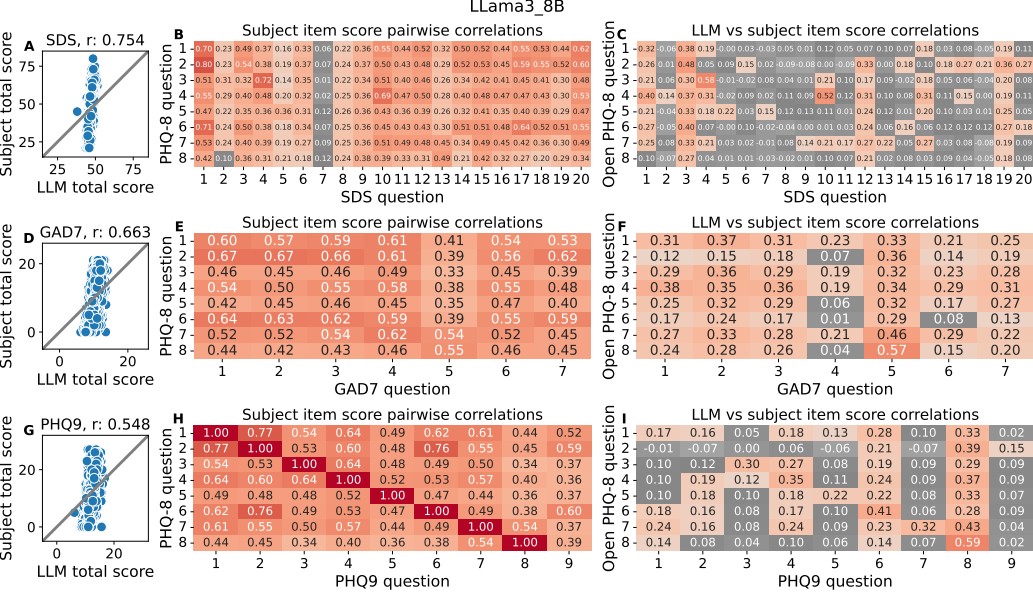

Figure 14: *Column I* (**A, D, G**): Subject's total scores for each questionnaire against the Llama3.1-8B total score estimate (Pearson correlation reported). *Column II* (**B, E, H**): Subjects' pairwise correlations between PHQ-9 item scores and SDS, GAD-7 and PHQ-9 scores. *Column III* (**C, F, I**): Correlations between participant's true item score for each questionnaire (x-axis) and Llama3.1-8B recovered score for that question given each of the open-ended PHQ-9 QA-pairs (y-axis). Significant correlations are in cool-warm colour. The correlations not reaching significance are in gray scale.

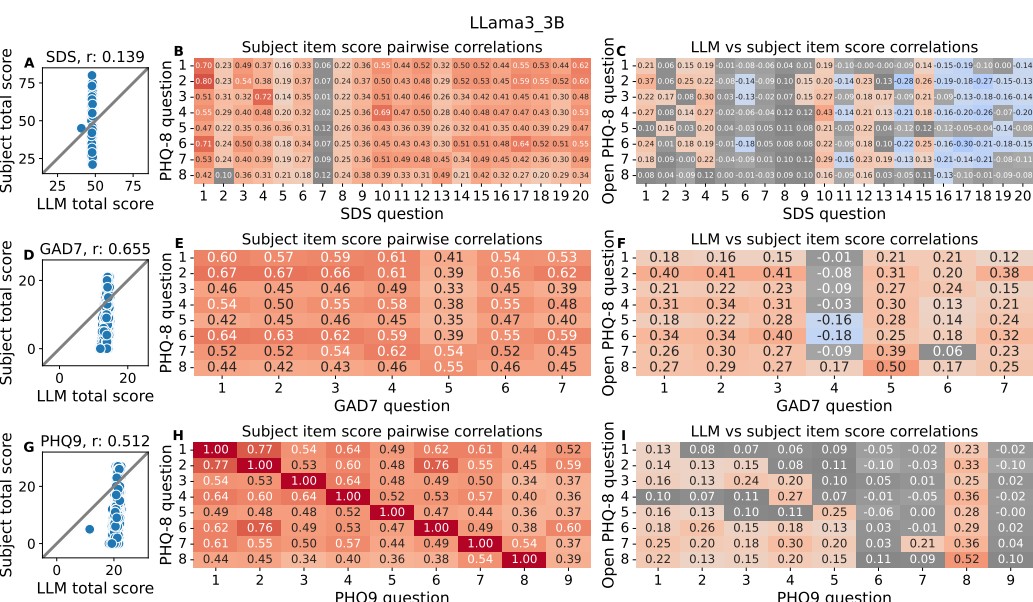

Figure 15: *Column I* (**A, D, G**): Subject's total scores for each questionnaire against the Llama3.2-3B total score estimate (Pearson correlation reported). *Column II* (**B, E, H**): Subjects' pairwise correlations between PHQ-9 item scores and SDS, GAD-7 and PHQ-9 scores. *Column III* (**C, F, I**): Correlations between participant's true item score for each questionnaire (x-axis) and Llama3.2-3B recovered score for that question given each of the open-ended PHQ-9 QA-pairs (y-axis). Significant correlations are in cool-warm colour. The correlations not reaching significance are in gray scale.

# C    EXTRACTING LATENT STRUCTURE WITH SUPERVISED SPARSE AUTO-ENCODER STUDY - STUDY 2

## C.1    SUPERVISED SPARSE AUTO-ENCODER (SSAE)

### C.1.1    ARCHITECTURE DETAILS

We outlined the supervised sparse auto-encoder equations below. We tied together the encoding and decoding matrix weights, the latter being normalised along the hidden dimension and transposed.

$$\boldsymbol{h}_{cent} = \boldsymbol{h} - \boldsymbol{b}_{dec} \tag{8}$$

$$\boldsymbol{s}_{post} = relu(\boldsymbol{h}_{cent}\boldsymbol{W}_{enc}) \tag{9}$$

$$\boldsymbol{q} = \boldsymbol{s}_{post}\boldsymbol{W}_{proj} + \boldsymbol{b}_{proj} \tag{10}$$

$$\boldsymbol{h}_{rec} = \boldsymbol{s}_{post}\boldsymbol{W}_{enc}^{T} + \boldsymbol{b}_{dec} \tag{11}$$

Here $\boldsymbol{W}_{enc}$ is $d \times (d * f)$, where $d$ is the original hidden state dimensionality from the LLM model $\boldsymbol{h}$, while $f$ is the dimension scaling factor.

We defined the following loss items, where $\boldsymbol{y}$ is the vector of participant scores ($Q$-dimensional, for PHQ-9 $Q = 9$)

Reconstruction loss:
$$\boldsymbol{l}_{rec} = ||(\boldsymbol{h}_{rec} - \boldsymbol{h})||^2 \tag{12}$$

Sparsity loss:
$$\boldsymbol{l}_{sparse} = ||\boldsymbol{s}_{post}||_1 \tag{13}$$

Question prediction loss:
$$\boldsymbol{l}_{qs} = ||(\boldsymbol{q} - \boldsymbol{y})||^2 \tag{14}$$

Questionnaire total severity loss:

$$\boldsymbol{l}_{sev} = ||\sum^{Q} q_j - \sum^{Q} y_j||^2 \tag{15}$$

Total loss

$$L_{total} = \boldsymbol{l}_{rec} + \lambda\boldsymbol{l}_{sparse} + \boldsymbol{l}_{qs} + \frac{1}{Q}\boldsymbol{l}_{sec} \tag{16}$$

### C.1.2    SSAE TRAINING

We extracted hidden-states on the token just after the open-ended response for each layer of the model from the middle layer onwards. We averaged the hidden states across questions obtaining an average hidden state for each participant, $\boldsymbol{h}_i$, we then matched this with the true vector of z-scored PHQ-9 scores $\boldsymbol{y}_i$. We split the 770 participant dateset into training, validation and test set (70%, 15%, 15%).

To find the optimal setting of hyper-parameters for the sSAE model Lee et al. (2025); Yun et al. (2021); Le et al. (2018), we search the grid for the following hyper-parameters.

- Optimiser learning rates: $\alpha \in [0.001, 0.0001]$
- SAE sparsity loss coefficient $\lambda \in [0.05, 0.1, 0.2]$
- SAE dimension scaling factor $f \in [1, 2, 4]$

We used a batch size of 32. With the training set, we used Kingma & Ba (2017) optimiser for each unique setting on the hyper-parameter grid (using back-propagation) to find the best network weights. Then, using the validation set, we found the best hyper-parameter setting for each layer based on the validation loss as well as the correlation between true and predicted questionnaire scores. The final results plotted in the paper are against the held-out test set.

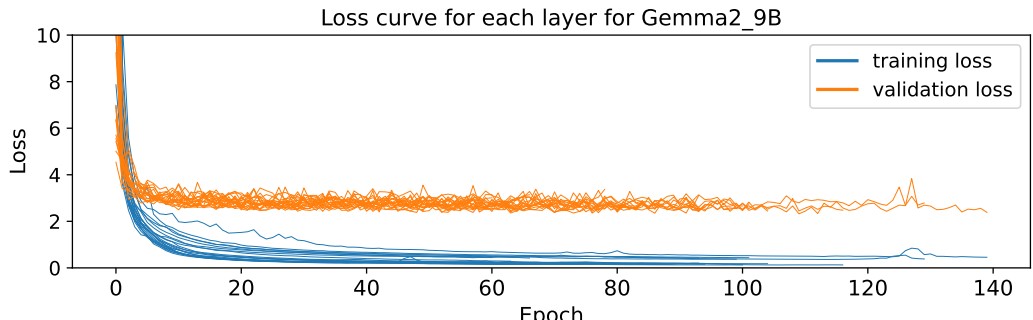

Figure 16: Plot of training and validation loss for the best hyper-parameter setting sSAE based on Gemma2-9B hidden state to predict PHQ-9 z-scores. Each line corresponds to layer-specific SAE.

### C.1.3  SSAE LATENT SCORE PERTURBATION

In the sSAE petrubation, we aim to achieve a specific change in question $\Delta q_j = [0,..,\delta_j,...,0]$ (here we always chose $\delta_j = 1$), such that $q_{new,j} = q + \Delta q_j$. Therefore we seek $\Delta s_{post,j}$, where

$$s_{new,j} = s_{post} + \Delta s_{post,j} \tag{17}$$

such that

$$W_{proj} s_{new,j} = q_{new,j} \tag{18}$$

leading to

$$W_{proj} \Delta s_{post,j} = \Delta q_j \tag{19}$$

We find $\Delta s_{post,j}$ by solving a basis pursuit problem (Van Den Berg & Friedlander, 2009) using `spgl1` python package (SPGL1, 2025). The goal is to minimise the following loss, encouraging sparse solutions.

$$\min \|W_{proj} \Delta s_{post,j} - \Delta q_j\|_1 \tag{20}$$

In the main paper, we then plot the perturbed predicted scores on the validation set as a confusion matrix.

### C.1.4  LOGITS STEERING USING PERTURBED SSAE LATENT STATES

Here we want to perturb the hidden states to change the severity of the multiple-choice responses given the open-ended response. For each question $j$, we extract the hidden states at each layer for the token just after the open-ended response and at the last token just before the response label is selected and average: $h_{avg,j}$. We then pass the $h_{avg,j}$ through the sSAE to perturb the latent state:$s_{post,avg,j} + \Delta s_{post_j}$, we then reconstruct this back to the original hidden-state space $h_{avg,pert,j}$.

At the last token of the input sequence, we alter the original hidden state $h_{last}$ sequentially, one layer at a time in the forward pass as follows. We first project the reconstructed perturbed sSAE state onto the original hidden state:

$$p = h_{last}\left(\frac{h_{avg,pert,j}}{\|h_{avg,pert,j}\|} \cdot h_{last}\right)\frac{1}{\|h_{last}\|} \tag{21}$$

/ and find the difference between the projection and the desired direction

$$v = h_{avg,pert,j} - p \tag{22}$$

We then add this difference to the original hidden state to move it in the desired direction with strength $\gamma$ (if the dot product was negative, we flip the direction of the hidden state first):

$$h_{last,pert} = sign\left(\frac{h_{avg,pert,j}}{\|h_{avg,pert,j}\|} \cdot h_{last}\right)h_{last} + \gamma v \tag{23}$$

Using the validation test we considered $\gamma \in [-1.5, -1, -0.5, -0.25, 0.25, 0.5, 1, 1.5]$. We looked for strongest Cohen-d effect sizes in the expected scores $\Delta y_{i,j} = \mathrm{E}[y_{i,j,pert} - y_{i,j}]$ and found that $\gamma \in [1.5, -0.25]$ provides the most robust changes. We then resampled the perturbed responses using the held-out test set, using these strength settings, and plotted the results in the main paper.

# D  GENERALISING TO EMOTIONAL INDUCTION - STUDY 3

## D.1  PARTICIPANT RECRUITMENT

### D.1.1  INCLUSION/EXCLUSION CRITERIA

In the open-ended questions about baseline mood and energy levels, as well as in the end of study positive re-evaluation, participants had to type at least 30 words in 100 seconds. In the recreate the audio task, participants had to reach 15 words per audio under 40 seconds. In the generate new diary participant had to reach 100 words under 3 minutes.

We excluded participants who meet at least one of the criteria:

- timed out or provided responses shorter than the required word limit above to any of the questions more than 4 times in total,
- failed two attention checks.

Moreover, given the auditory nature of the experiment, we included participants that

- reported no dyslexia, speech disorders or hearing difficulties
- not wearing a cochlear implant
- passed a technical test establishing whether their speakers/headphones are working correctly

In the emotional induction analyses we excluded subject outliers, whose PHQ-9 Q2 score changes were beyond the 3 standard deviations from the mean.

In total we included 190 participants.

### D.1.2  DEPRESSION SEVERITY

We report he summary statistics of total scores for PHQ-9, GAD-7, SDS below:

- PHQ-9 total $11.85 \pm 5.44$ (depression)

## D.2  MEASURES

### D.2.1  OPEN-ENDED QUESTIONS

At baseline and after mood induction, we asked partiicpant the following questions that they had to asnwer in an open-ended manner, similarly as in Study 1. They had to write a minimum of 30 words under 100 seconds.

1. For the past two weeks, have you been bothered by your mood and how you felt generally? Were there any situations when you felt down, depressed, or hopeless?

2. Have you been bothered by your energy levels over the past two weeks? Can you recall situations when it comes to feeling tired/lively or low/high on energy?

### D.2.2  MOMENTARY MOOD

Before and after mood induction, participants were asked to rate their momentary mood by answering a question "How happy are you at this moment?" on a continuos visual analogue scale ranging labels: Very unhappy, Quite unhappy, A bit unhappy, Neutral, A bit happy, Quite happy, Very happy. See Fig 17.

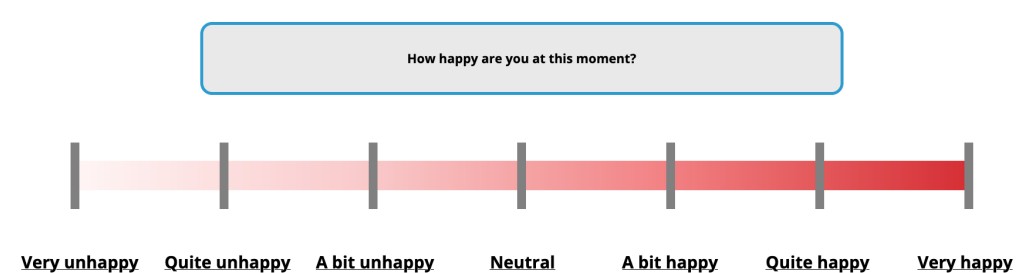

Figure 17: Screenshot of the mood rating VAS scale used in Study 3.

### D.2.3 PHQ-9 VAS

Before and after mood induction, participants were asked PHQ-9 items on a visual analogue scale ranging the labels: Not at all, Several days, More than half the days, Nearly every day. See Fig 18.

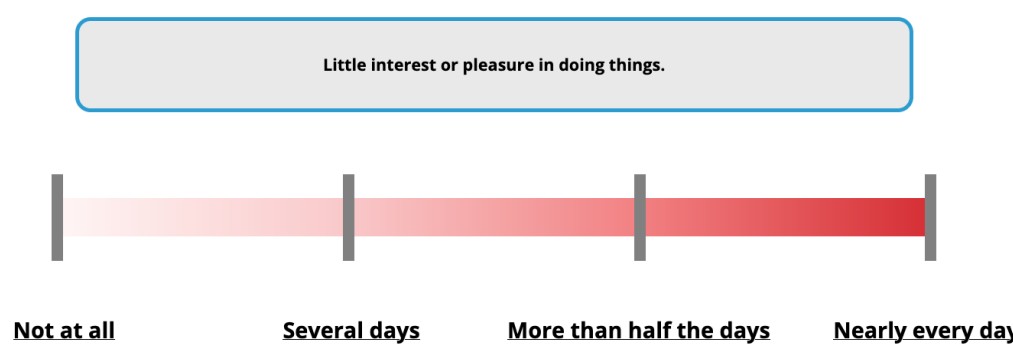

Figure 18: Screenshot of the PHQ-9 rating VAS scale used in Study 3 for example question

### D.2.4 WORD RECALL

Before the mood induction, participants were asked twice to observe words being displayed one at a time (in random order) and then recall as many as possible in 45 seconds. We used the following list of words:

VIGOROUS, ENERGETIC, LIVELY, EXHAUSTED, TIRED, DRAINED, JOYFUL, DELIGHTED, HAPPY, UNHAPPY, HOPELESS, MISERABLE, GENUINE, WHOLESOME, ETHICAL, CORRUPT, SLOPPY, UNSAFE.

After the mood induction, we asked them to recall the words again. We used SiEBERT - English-Language Sentiment Classification model (Hartmann et al., 2023) to calculate average sentiment scores of the recalled words before and after mood induction.

### D.3 INTERVENTION SELECTION

#### D.3.1 IDENTIFYING MOOD INDUCTION PROMPTS

To identify mood-induction diaries, we take all participants' open-ended responses (from study 1) and focus on PHQ-9 mood/hopelessness question (Q2). For each open-ended response, we have true subject score and the expected llm-predicted score. We also calculate the absolute difference

between subject and llm score. We filter open-ended responses where the absolute difference is $<= 0.75$ and where participant wrote at least 30 words. We then select open-ended responses where participants provided score 0 and score 3 (two extremes). We select the most extreme ones and edit the entries to create four entries ranging 33-59 words each (182-192 words total), aiming to maintain the content and severity of the original responses, representative of the average mental state of interest (low/high mood levels).

### D.3.2 AUDIO GENERATION

To generate the audio version of these diary entries, we use platform HumeAI (2025) platform, that offers a text-to-speech model (Octave), sensitive to the meaning of words in context. The model additionally allows for control over the level of expressiveness and other prosodic features. In our case, to control for speech prosody and emotional expressiveness across conditions (keeping these as constant as possible), we designed a custom voice with the following voice prompt:

```
A female voice, standard British English (Modern RP). Prosody: Low pitch
    variation with a flat, monotone-like contour. Pace: Slow and
    deliberate, approximately 150 words per minute, with natural pauses.
    Articulation: Clear and precise but with a casual, soft-spoken
    quality, avoiding harsh sibilance or plosives. Affect: Consistently
    neutral and emotionally detached.
```

We report the transcripts of the interventions below.

### D.3.3 POSITIVE MOOD INDUCTION TRANSCRIPTS

1. Lately, I've been really focused on making good memories and enjoying happy moments with my friends and family. I feel joyful, grateful, and full of positive energy. I think it's because things have been going my way lately — it's got me feeling excited and motivated.

2. I was feeling elated today — there was so much positivity around me. I'm surrounded by people I love, and that really makes me happy. Work's been great too; I have awesome colleagues, and we just wrapped up a big project together. I'm excited for what's next and ready to keep achieving more.

3. I've been in a happy, light mood and feeling really in control. I'm going out more freely now, with no anxiety, and actually enjoying hanging out with friends. I've just been feeling good doing all the regular life stuff. Overall, I'm optimistic, content, and hopeful about what's ahead.

4. I've had a really positive couple of weeks, with good things happening both at work and at home. I also took some time off to focus on myself and just relax. It's such a nice feeling — I just feel happy and content.

### D.3.4 NEGATIVE MOOD INDUCTION TRANSCRIPTS

1. I've been feeling really overwhelmed lately. It's been tough — I've felt lonely and depressed, like something bad is hanging over me. I can't even remember the last time I didn't feel this way.

2. I feel like I'm wasting my life. It's hard to enjoy anything. Everything just seems pointless, and I don't know how to change it. I'm so tired of getting rejection after rejection with job applications. I keep comparing myself to others, and it makes me feel pathetic. The world just feels like a cold, evil place.

3. I just feel too low to do anything. At work, all I can think about is getting home, and once I'm home, I just want to sleep the day away. Most nights I just vape some weed to try and chill, but the feelings don't really go away — they're just always there.

4. My birthday was supposed to be a happy time with family, but I just felt hollow and empty. It was like this wave of sadness hit me, and it hasn't gone away since. I feel so hopeless, and even the things I used to love don't bring me any happiness anymore.

### D.4 INSTRUCTIONS AND QUESTIONS

We report the exact instructions given to participants when responding in the study.

#### D.4.1 RECREATE INSTRUCTIONS

```
Next, you will listen to diary entries from the same person.
As you listen, put yourself in their shoes and imagine you are an actor
    playing their character.
-
You will need to reenact each entry by typing what you've heard.

Be as detailed as possible, but don't worry about typos or memorising
    word by word.

Focus on richly capturing the gist from a first-person perspective.
```

#### D.4.2 CREATE INSTRUCTIONS

```
Now, think about your own life.
Recall situations and events that might have been similar to the ones you
    've just heard and described.
Imagine you had to write a diary entry to describe how you felt and what
    you thought about.
What would you write?
Be as detailed as possible!

If you've exhausted one train of thought, start another.

Keep typing until the time runs out.
```

#### D.4.3 POSITIVE RE-EVALUATION INSTRUCTIONS

These were asked at the end of the study, the goal was to move participants into the positive mental state by positively re-evaluating their baseline open-ended responses.

```
Revisit both your responses from before about your mood, feelings and
    energy levels. Now, reframe them more positively and rewrite them
    here. Remember to give examples of situations or experiences that
    best illustrate these
```

## D.5 SAE RESULTS

### D.5.1 SAE SCORES MOOD INDUCTION

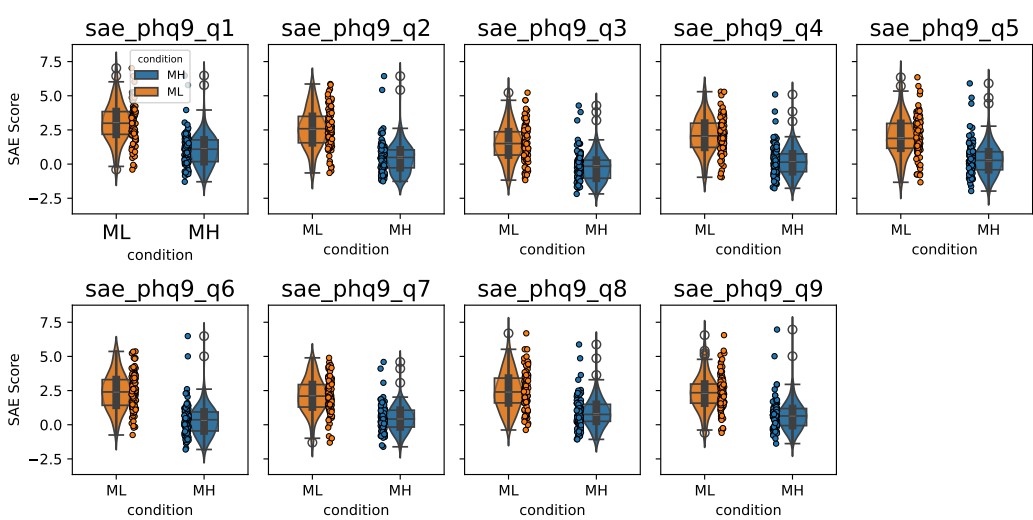

Figure 19: sSAE latent scores for each participant's created, autobiographical diary entry show the sSAE measure significantly differentiates the inducing emotional state of each condition

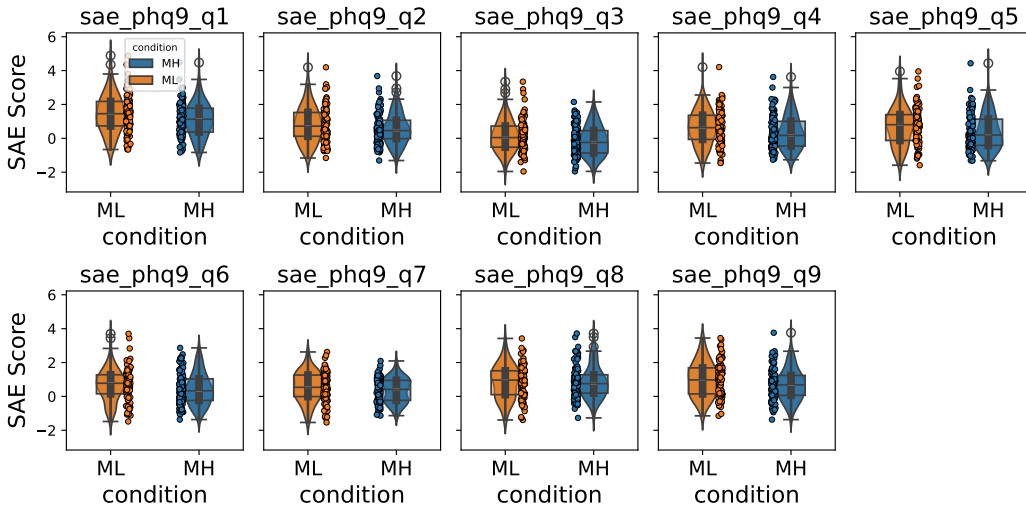

Figure 20: sSAE latent scores for each participant's positive re-evaluations open-ended descriptions.

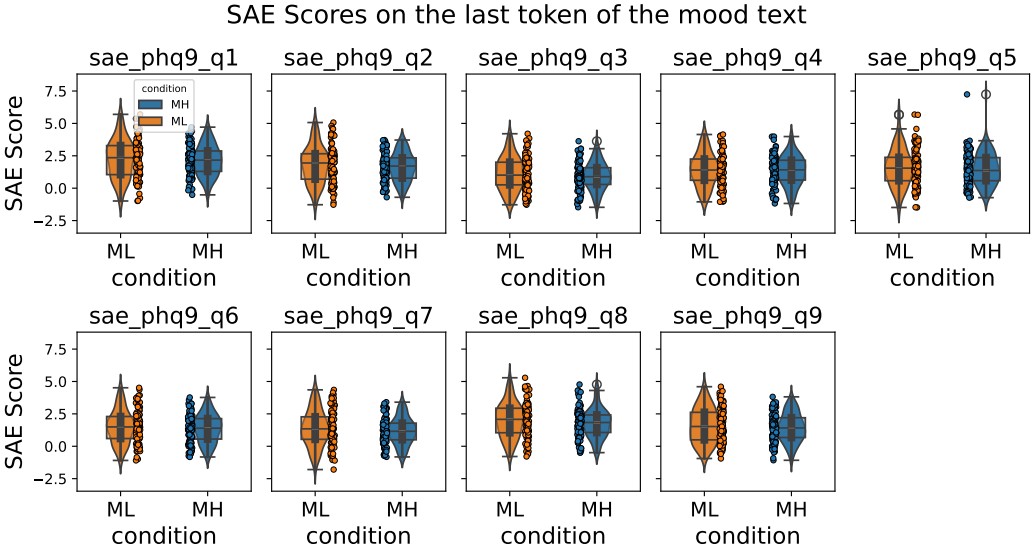

Figure 21: sSAE latent scores for each participant's baseline mood open-ended descriptions.

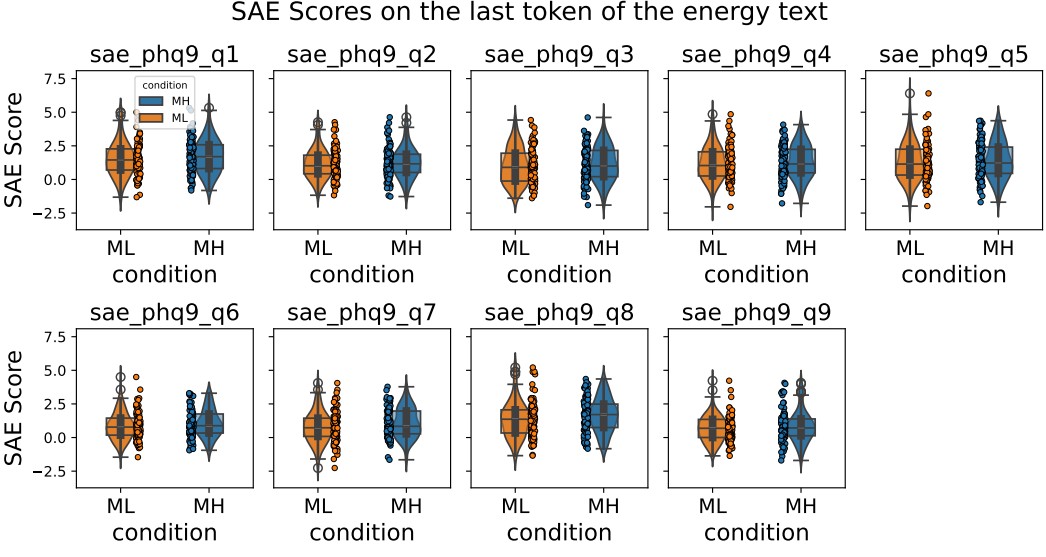

Figure 22: sSAE latent scores for each participant's baseline energy levels open-ended descriptions.

# E  MISCELLANEOUS

## E.1  PACKAGES

We list some of the packages and versions we used across the experiments:

- `transformers v4.48.0`
- `scikit-learn v1.7.1`
- `scipy v1.16.1`
- `pandas v2.2.0/v2.3.1`
- `numpy v1.2.6/v2.3.2`
- `seaborn v0.13.2`
- `baukit v0.0.1`
- `einops v0.8.0`
- Logit sampling `python v3.9.18`
- sSAE sampling `python v3.13.5`
- sSAE hyperparameter search, logit sampling and perturbation `torch v2.2.0+cu121`
- sSAE evaluation and best model fitting torch v2.6.0

## E.2  LLMs DETAILS

We used the Gemma2 2b-it and 9b-it model version GemmaTeam (2024), as well as Llama-3.1-8B-Instruct and Llama-3.2-3B-Instruct (AI@Meta, 2024). We also used Mistral-7B-OpenOrca large language model Lian et al. (2023). The model is a fine-tuned version of Mistral-7B-v0.1 model (Jiang et al., 2023). The original Mistral model was finetuned on a rich collection of augmented FLAN data aligns (Longpre et al., 2023) based on Orca model (Mukherjee et al., 2023). Essentially the model was fine-tuned on rich signals from GPT-4 model (OpenAI, 2023) that include step-by-step thought processes and complex instructions guided by teacher assistance.

