# OpenReview forum: "Quantifying depressive mental states with large language models"
_ICLR.cc/2026/Conference — ICLR 2026 Conference Withdrawn Submission_

### Official Review · Reviewer_dkn7 · 2025-10-22

**Soundness:** 3
**Presentation:** 3
**Contribution:** 1
**Rating:** 2
**Confidence:** 4

**Summary:**

This paper presents a three-part framework for evaluating the ability of large language models (LLMs) to quantify depressive symptoms from open-ended verbal descriptions. The authors introduce a novel dataset combining open-ended responses with clinically validated depression measures, propose a supervised sparse auto-encoder (sSAE) to extract latent symptom representations, and validate the approach using an emotion induction experiment. While the topic is timely and the experimental design is ambitious, the paper suffers from significant methodological, conceptual, and ethical shortcomings that undermine its contributions.

**Strengths:**

The collection of paired open-ended and standardized questionnaire data is a valuable contribution to the field. The three-test framework is conceptually sound and addresses different aspects of model performance.

 The use of mood induction to test model sensitivity to emotional change is a strong experimental design choice.

 The use of sSAEs and perturbation analyses demonstrates a sophisticated approach to interpreting latent representations.

**Weaknesses:**

The paper conflates verbal expression with mental state. LLMs are trained on language, not mental states, and the leap from text-based prediction to mental state quantification is not sufficiently justified. The authors claim LLMs "encode a consistent representation akin to a mental state," but this is speculative and not supported by causal or theoretical grounding.

In Study 3, the mood induction stimuli were derived from Study 1 responses, which were already used to train the sSAE. This introduces a risk of overfitting and limits the generalizability of the results.  The use of LLM-predicted scores to select mood induction prompts (Appendix D.3.1) may bias the intervention toward model-friendly examples.

Some new LLM-based mental health methods are not referred, such as mentalllama. The 1.5-minute time limit and 30-word minimum for open-ended responses may have influenced response quality and model performance in unaccounted ways. The authors suggest potential applications in "therapeutic intervention allocation and delivery" and suicide risk identification, but provide no evidence that the model is safe, reliable, or clinically valid for such high-stakes use.

Model performance, while "moderate to strong," is evaluated only on a specific online population with depressive symptoms. There is no evidence that results generalize to clinical populations, other cultures, or non-English speakers.

**Questions:**

How do you justify the claim that LLMs capture mental states as opposed to linguistic patterns associated with self-reported symptoms?

How do you rule out the possibility that the sSAE’s sensitivity to mood induction is an artifact of using LLM-scored prompts from the same dataset used for training?

Can you provide any evidence that the sSAE latent dimensions correspond to clinically meaningful constructs beyond symptom severity?

---

> ### Author Response · Authors · 2025-11-20
>
> We are grateful for the insightful comments and time spent on reviewing our submission. Please find our responses below.
>
> We argue that a verbal expression is part of the mental state, and for the analyses we assume the verbalisation is an informative proxy for the underlying mental state. Our argument stems from the observations regarding standardised questionnaire responses that encode a very consistent pattern, shown to be indicative of stable constructs related to depression, anxiety, OCD and beyond (for example by factor analysis). We expand this argument to open-ended responses and show that the structure is preserved when verbalisations are represented by LLM states.
>
> Thank you for this valid concern. We select the prompts based on both LLM and true participant scores. Indeed, while, theoretically, there may be some leakage, we hope that this was minimised as the input to the sSAE in Study 3 is based on participants continuations of that prompt rather than the reproduction. We are open to suggestions as to how best show that the leakage is minimal.
>
> Thank you for pointing out mentallama, we will situate our work better in the future. We agree that time and word constraints might affect the quality, something we hope to alleviate with a large sample size. We hope that this study informs future endeavours across variety of settings and can inspire application, rather than propose our approach specifically (hence no in-depth safety, reliability and validity analyses, which we agree are important).
>
> Online population is a good starting point, but as you point out not fully representative of the clinical population and other cultures. We hope that this study informs future endeavours across variety of settings.

---

### Official Review · Reviewer_fVTX · 2025-11-01

**Soundness:** 2
**Presentation:** 2
**Contribution:** 2
**Rating:** 4
**Confidence:** 3

**Summary:**

This paper proposes a framework to evaluate the capacity of Large Language Models to quantify depressive mental states. The work tests model performance against human data, extracts a latent representation of symptoms, and assesses the representation's sensitivity to induced changes in emotion, which provides a method for validating language models for mental health applications.

**Strengths:**

The use of three tests provides a framework for model evaluation. The first test establishes a performance ceiling using a dataset that links verbal descriptions with clinical scores. The second test moves from performance to mechanism by training a sparse auto-encoder to find a latent structure. The perturbation and steering analyses show a connection between the latent space and model output. The third test connects the model's internal states to human psychological states through an emotion induction experiment. this sequence of performance, mechanism, and external validity provides a systematic approach to the problem.

**Weaknesses:**

The open-ended text from participants was constrained by time and word count, which may influence the quality of the language data the models received. The population was recruited from an online platform, which may not represent the general population or clinical populations. The emotion induction focused on one dimension of affect; the sensitivity of the derived measures to other states of emotion is not known. The paper equates model performance on questionnaire prediction with quantification of a mental state, which is a step in reasoning that requires more justification.

**Questions:**

How might the sSAE's latent structure differ if trained on data from a clinical population undergoing treatment, where symptom expression might change over time?

The steering intervention modifies the hidden state. What is the effect of this modification on the semantic content of text generated from that perturbed state? Does it produce text that reflects the steered symptom severity?

Could the framework be adapted to quantify other mental constructs, such as anxiety or psychosis, and what challenges would be anticipated?

---

> ### Author Response · Authors · 2025-11-20
>
> We are grateful for the insightful comments and time spent on reviewing our submission. Please find our responses below.
>
> In this work, we are interested in establishing and evaluating an LLM approach towards quantifying symptoms from verbalisations. We agree that time and word constraints might affect the quality, something we hope to alleviate with a large sample size. Online population is a good starting point, but as you point out not fully representative of the clinical population. We hope that this study informs future endeavours across variety of settings. Moreover, we are considering expanding the interventions to capture fine-grained states, for example energy levels.
>
> We agree this is would be a very interesting question to purse, and we hope an approach like ours will be applied in clinical population and treatment understanding. It would be important to establish the reliability of such measures over time as well.
>
> We haven’t sampled forms of texts other than the questionnaire response. This would an interesting way to try to model changes is affective states measures through rich text, something we are interested in pursuing in the future (for example starting with our mood-induction dataset).
>
> Indeed, we hope that this approach could be expanded to other syndromes. See some related work in schizophrenia - https://polytechnique.hal.science/hal-05086512/

---

### Official Review · Reviewer_LQco · 2025-11-01

**Soundness:** 2
**Presentation:** 1
**Contribution:** 2
**Rating:** 2
**Confidence:** 3

**Summary:**

The paper aims to evaluate LLMs performance on 3 tests to quantify depressive mental states. They recruited human participants with diverse levels of depression to gather responses from mental health questionnaires (PHQ, GAD, SDS) as data for the LLMs.
For test 1, they attempted to predict item-level multiple-choice question scores based on the participants' item-level open-ended text responses.
For test 2, they use ground-truth from test 1 to train a supervised sparse auto-encoders (sSAE) to predict the participants’ z-scored PHQ-9 scores from the hidden states representations of the best-performing LLM from test 1.
For test 3, they evaluated whether the sSAE measures from test 2 can capture and quantify specific emotional changes induced through text.

**Strengths:**

Originality:
The authors recruited participants (770 for test 1 and 190 for test 3) to collect real-life data for their experiments. This inherently separates their analysis and results from other studies in similar applications, which provides new and fresh insights.
I am not certain if their methodology to evaluate LLMs in this way is novel.


Quality:
A comprehensive study was done on this topic with 3 tests conducted while building upon the previous.


Clarity:
The authors provided detailed explanation of the methods and implementation.


Significance:
N/A

**Weaknesses:**

The authors did not provide sufficient explanation for their choice of plots and graphs to demonstrate their evaluation results, such as figures 2 and 3.
It is unclear what some of these figures are meant to show and the authors offer limited insights and descriptions from the visualization.

The authors offer limited insights into the reason behind the LLMs performance in these tasks despite claiming that they aim to describe the LLMs' promises and limitations (i.e. "The LLMs performed well/poorly in this 'abc' task because of 'xyz' possible reason.").

The authors did not offer suggestions as to how the results of their study can translate to a real-life application of LLM in mental welfare.

They did not discuss much about prior related work that also attempted to use LLMs for depressive symptoms detection for a comparison.
This makes it harder to judge whether their work is significant within this area.

**Questions:**

Typos:
Line 367: "continuos"
Line 1329: "he"
Line 1337: "partiicpant"



Formatting Issues:
Authors made LaTeX formatting error. Refer to section 3.1.1. For quotation marks, using regular double quotes "" is incorrect. ``'' is needed.
The graphs in B.5.2 need to be adjusted. The letters overlap with the top number.



Questions:
1. I do not have expertise on scores such as PHQ, GAD, or SDS but I believe that the statistics shown in Appendix B.1.2 does not indicate that the 770 human sample show a very prominent level of depression.
Is this a regular population sample where majority of people are not depressed? If so, this could affect the validity of the evaluation for depressive symptoms.
Please correct me if my understanding of the scores are incorrect.

2. Are there any overlap between the participants in study 1 and 3? If some participants overlap, then this could create some potential bias in the training data for study 3 and cause the result to appear better.

3. Should the authors perform comparisons between groups such as non-depressed or happy participants and severely depressed participants to see how the LLMs would behave?

4. In Figure 2I, is there a reason why performance is noticeably worse for Q1 and Q8?
I did not see a possible explanation or hypothesis mentioned for this.

5. For study 3, is the analysis comprehensive enough to only include positive (mood high) and negative (mood low) emotional states from the participants?

6. Are these experiments truly enough to establish a baseline / ceiling / limitation for LLMs ability to quantify depressive mental states?
It is difficult to see how significant your results are in this area without a comparison with other studies that also attempted to find evaluation methods for depressive signs with LLMs.

---

> ### Author Response · Authors · 2025-11-20
>
> We are grateful for the insightful comments and time spent on reviewing our submission. Please find our responses below.
>
> Thank you for highlighting this, we will work on the better presentation, motivation and justification in the future iteration of this work. We agree that it’s crucial to better highlight the applications that could stem from this work.
>
> Thank you for catching the formatting issues!
>
> The severity of depression in the sample is moderate to moderately-severe with some severe and mild cases based on PHQ-9 cut-offs. We will highlight and clarify this, and include a histogram in future iterations.
>
> No, the samples are non-overlapping in this instance. We screened-out participants who took part in the first study.
>
> Thank you for the suggestions. This would be a good extension to consider other populations, which we are considering in the future.
>
> Indeed, we haven’t discussed this in the paper much. We suspect this might be down to how we phrased the question which affected the quality of the produced participant responses used by LLMs to predict scores. Both start with ‘could/can you share/describe…’, which may bias towards shorter and less detailed responses compared with other questions.
>
> We are considering expanding the interventions to capture fine-grained states, for example energy levels.
>
> Thank you for highlighting this. Our approach is to use large human dataset itself as that which establishes the limit and compare how LLMs fare against this upper bound. We agree that there is other relevant work and we will aim to flesh out a comparison to our work.

---

### Official Review · Reviewer_hNP2 · 2025-11-02

**Soundness:** 2
**Presentation:** 2
**Contribution:** 2
**Rating:** 2
**Confidence:** 3

**Summary:**

This paper studies whether off-the-shelf large language models (LLMs) encode clinically meaningful information in their hidden representations that correspond to depressive and emotional mental states. The authors propose a three-part evaluation framework to determine the potential and limitations of LLMs for quantifying mental health symptoms using data from 770 participants who provided open-ended text responses alongside standard depression and anxiety questionnaires (PHQ-9, GAD-7, SDS).

First, they show that several instruction-tuned models (Gemma2, Llama3, Mistral) can predict questionnaire scores using the answers to open-ended questions as their context. Second, they train a supervised sparse autoencoder (sSAE) on LLM hidden states to model the latent structure of depressive symptoms, demonstrating selective control over symptom dimensions via latent perturbations. Finally, in a mood-induction experiment with 190 participants, the sSAE latent scores track experimentally induced emotional changes.

Based on these results, the paper concludes that LLMs encode structured representations of depressive mental states that can be extracted and manipulated through supervised latent modeling.

**Strengths:**

- The paper introduces a large dataset on mental disorders. The dataset contains rich ground truth for mental health modeling, consisting of matched open-ended Q&As and several questionnaires.
- The paper presents evidence for all experiments with statistical rigor.

**Weaknesses:**

- The paper contributions lie in its dataset and experimental design related to mental health. However, the paper offers limited novelty in its methods for quantifying or analyzing the LLM states.
- While the results are promising, the use of a supervised sparse autoencoder (sSAE) in Study 2 (and 3) provides limited theoretical insight into the nature of LLM state representations. The observed predictive performance may primarily reflect the supervised model’s ability to fit symptom scores rather than evidence that the LLM itself encodes meaningful mental-state dimensions. It would be more informative if the authors could demonstrate direct manipulation of the LLM’s own outputs or internal activations based on the discovered sparse coding, rather than relying on a separate probing model.
- The writing occasionally overstates the contribution or uses vague phrasing. For example, it is unclear what is meant by “further establishing the sensitivity and limits of LLM quantification” (Line 437). Phrases like this imply a theoretical characterization of model capability, but the paper only presents empirical correlations on a specific dataset.  Terms such as “upper bound,” “hard limits,” and “conceptual alignment” are used without precise definition or analytical support.
- The third experiment aims to test whether LLM-derived latent measures track participants’ emotional changes after a mood induction. However, it is unclear how the emotion information is actually represented or provided to the LLM.

**Questions:**

1. How exactly was the emotional state “input” to the LLM?
2. In C.1.4, why was the average between the two hidden states appropriate? Why not in C.1.2?

**Details Of Ethics Concerns:**

The authors state that they obtained approval from an ethical committee.

---

> ### Author Response · Authors · 2025-11-20
>
> We are grateful for the insightful comments and time spent on reviewing our submission. Please find our responses below.
>
> Thank you for highlighting this, we agree that perhaps the novelty regarding analysis of LLM states is lacking. However, our goal was to present and  evaluate LLMs as a tool to capture and quantify human mental states measured through symptom verbalisations. Perhaps we failed to convey the emphasis on the human side of the story, which is something we will change in future submissions.
>
> We perturb the internal activations and then sample scores
>
> We will consider rephrasing to capture the limited scope of the empirical argument.
>
> In the first study, we use the open-ended prompt as described in the appendix B.4. In the mood-induction study, we apply the trained sSAE model to the Gemma2-9b embeddings (last token, focusing on the previously established best-performing layer) of diary texts that participants have written and then read out the latent PHQ-9 (depression symptoms) sSAE score vectors. In particular, we focus on Q2 score as this was the target of our intervention. In our future manuscripts, we will endeavour to better explain this.
>
> In C.1.2, we extracted hidden states of the text that describes the symptoms (this is before them multiple-choice question part of the prompt) at the last token of the description. Using the training set, we found that this approach didn’t produce consistent logit changes in the perturbation analysis, perhaps due to the fact that the change in the open-ended part of the prompt is not “represented enough” at the later multiple-choice part. We therefore decided to take an average across two tokens (after open-ended response, and the final token) to capture both the symptom change and the question-answering ability, and perturb this averaged representation. Any evaluations after this were done on a held-out test set.

---

### Note · Authors · 2025-11-20

**Comment:**

Again, we are thankful for reviewers' feedback. We have provided our responses. However, after taking their comments into account, we decided to withdraw the paper.

**Withdrawal Confirmation:**

I have read and agree with the venue's withdrawal policy on behalf of myself and my co-authors.